# Using the International Tree-Ring Data Bank (ITRDB) records as century-long benchmarks for global land-surface models

Jina Jeong[1], Jonathan Barichivich[2,3] , Philippe Peylin[2], Vanessa Haverd[†,4], Matthew J. McGrath[2], Nicolas Vuichard[2], Michael N. Evans[5], Flurin Babst[6,7,8], and Sebastiaan Luyssaert[1]

[1]Department of Ecological Sciences, VU University, 1081HV Amsterdam, the Netherlands.
[2]Laboratoire des Sciences du Climat et de l'Environnement, IPSL, CNRS/CEA/UVSQ, 91191 Gif sur Yvette, France.
[3]Instituto de Conservación Biodiversidad y Territorio, Universidad Austral de Chile, 5090000 Valdivia, Chile.
[4]CSIRO Oceans and Atmosphere, Canberra, 2601, Australia.
[5]Department of Geology & ESSIC, University of Maryland, MD 20742-4211, USA.
[6]Dendro Sciences Group, Swiss Federal Research Institute WSL, Zürcherstrasse 111, CH-8903 Birmensdorf, Switzerland.
[7]School of Natural Resources and the Environment, University of Arizona, Tucson, USA.
[8]Laboratory of Tree-Ring Research, University of Arizona, Tucson, USA.
[†]Deceased 29 January, 2021.

**Correspondence:** Jina Jeong (j.jeong@vu.nl)

**Abstract.**

The search for a long-term benchmark for land-surface models (LSM) has brought tree-ring data to the attention of the land-surface modeling community, as tree-ring data have recorded growth well before human-induced environmental changes became important. We propose and evaluate an improved conceptual framework of when and how tree-ring data may, despite their sampling biases, be used as century-long hindcasting targets for evaluating LSMs. Four complementary benchmarks – size-related diameter growth, diameter increment of mature trees, diameter increment of young trees, and the response of tree growth to extreme events – were simulated using the LSM ORCHIDEE version r5698, and verified against observations from 11 sites in the independent, unbiased European biomass network datasets. The potential for big-tree selection bias in the International Tree-Ring Data Bank (ITRDB) was investigated by subsampling the 11 sites from European biomass network . We find that in about 95% of the test cases, using ITRDB data would result in the same conclusions as using the European biomass network when the LSM is benchmarked against the annual radial growth during extreme climate years. The ITRDB data can be used with 70% confidence when benchmarked against the annual radial growth of mature trees or the size-related trend in annual radial growth. Care should be taken when using the ITRDB data to benchmark the annual radial growth of young trees, as only 50% of the test cases were consistent with the results from the European biomass network. The proposed maximum tree diameter and annual growth increment benchmarks may enable use of ITRDB data for large-scale validation of LSM-simulated response of forest ecosystems to the transition from pre-industrial to present-day environmental conditions over the past century. The results also suggest ways in which tree-ring width observations may be collected and/or reprocessed to provide long-term validation tests for land surface models.

20   **Running head:** Tree-ring records as century-long benchmarks

  **Key words:** forest growth, tree-ring width, diameter growth, climate sensitivity, size-dependent growth, climate change

# 1 Introduction

Earth system models integrate numerical submodels of atmospheric circulation, ocean dynamics and biogeochemistry, sea ice dynamics, and biophysical and biogeochemical processes at the land-surface. Climate projections made by Earth system models have been the corner-stone of the all Assessment Reports of the Intergovernmental Panel on Climate Change (IPCC, 2013) and as such have made a tremendous impact on global environmental policy (Paris Agreements, 2015). The credibility of projections of the future climate from any Earth system model in part relies on the ability of each of its four submodels to accurately reproduce the past (McGuffie A. and Henderson-Sellers, 2005). Although long-term changes that date back to pre-industrial conditions (Luo et al., 2012) have been documented for vegetation distribution through pollen based reconstructions (Cao et al., 2019), land-surface models (LSMs) currently lack a long-term benchmark for forest ecosystem functioning. The absence of long-term benchmarks is thought to contribute substantially to uncertainties in simulated future global carbon stocks in soil and vegetation (Friedlingstein et al., 2006, 2014) and as such to climate projections (Fig. S1a).

Tree-ring records provide annual information on historical tree growth and physiology in relation to environmental conditions, including during the time before human activities started to affect the atmospheric $CO_2$ concentration (Fritts, 2012; Hemming et al., 2001). Even though trees grown in the absence of a clear annual rhythm of vegetative and dormant seasons may not develop distinct tree rings, as observed for many species from the humid tropics, tree-ring records have been proposed as a large-scale and long-term benchmark for the land surface component of Earth system models (Fig. S1b; See section 6 for more details) (Babst et al., 2014a, b, 2017, 2018; Zuidema et al., 2018).

Until now, tree-ring records have often been collected to reconstruct past climate and hydrological variability from sites where trees grow near the colder or drier fringes of their distribution (Briffa et al., 2004; D'Arrigo et al., 2008). The most comprehensive archive of publicly shared tree-ring data is the International Tree-ring Data Bank (ITRDB), with more than 4,000 locations from 226 species across most forested biomes (Fig. S2) (Grissino-Mayer and Fritts, 1997; Zhao et al., 2019). However, a shortage of site metadata and the prevailing geographical, species and tree selection sampling biases resulting from targeting climate-sensitive trees has limited the use of the ITRDB archive to infer long-term changes in forest growth (Bowman et al., 2013; Briffa and Melvin, 2011; Klesse et al., 2018; Zhao et al., 2019). Compared to tree-ring records that were collected for the purpose of benchmarking LSMs, such as the European tree-ring network of biomass plots (hereafter called "European biomass network"; Klesse et al. (2018)) that is available through the database of the BACI project (BACI, 2020), the aforementioned issues may limit the information content of the ITRDB records. This incomplete information content should, however, be balanced against the associated benefits in terms of time gain and resource savings when re-using the large ITRDB dataset.

If tree rings are to be used as benchmarks for LSMs, the models must demonstrate skillful simulations of tree-ring width (TRW). In the past decades, the major physiological and ecological processes that are responsible for annual tree-ring growth became sufficiently well-understood to be formalized in mathematical models with different levels of details. The first TRW models (Wilson and Howard, 1968) described processes at the cell level: cell division, cell enlargement, and cell wall thickening. Later,

the carbon and water balance of trees was added (Fritts et al., 1999) as well as a parameterization of the influences of climate on cambial activity (Vaganov et al., 2006). These models were capable of reproducing short-term radial growth at the tree level. Further developments introduced a notion of turgor and hormone regulation for cell growth (Drew et al., 2010; Hölttä et al., 2006; Leuzinger et al., 2013; De Schepper and Steppe, 2010; Steppe et al., 2006).

At the same time, the spatial scale of models simulating wood formation based on cell dynamics was extended to the stand
level by simplifying the representation of processes. In these models, photosynthate availability, air temperature and soil water content were used to constrain wood cell growth and successfully reproduced observations (Deleuze and Houllier, 1998; Hayat et al., 2017; Wilkinson et al., 2015). Further simplifications were proposed by simulating the radial growth of trees based solely on carbon allocation (Deleuze et al., 2004; Merganičová et al., 2019) rather than cell dynamics, the latter being computationally too expensive for large scale vegetation models (Li et al., 2014; Misson, 2004; Sato et al., 2007). Hence, a
variety of approaches is now available to describe TRW growth in forest models, dynamic vegetation models and LSMs, but to the best of our knowledge there is yet no land-surface component of any Earth system model with such capability.

This study articulates an improved conceptual framework for benchmarking simulated radial growth against ITRDB tree-ring data, addressing limitations in the models, the data and the methods to compare models and data. The aims are to: (1) use current understanding of tree-ring growth to derive the minimal requirements for benchmarking LSMs against tree-ring
records archived in the ITRDB; (2) review potential issues of using the ITRDB to benchmark LSMs; (3) propose solutions for a meaningful comparison of LSMs against ITRDB records; and (4) verify the proposed solutions by benchmarking a LSM using data from a European biomass network (BACI, 2020) that is not prone to sampling biases related to dendroclimatic research. The organization of this paper follows these aims, and dependencies between these aims and the workflow of this study are detailed in Fig. 1.

## 2 Background: model requirements, data limitations and benchmarks

### 2.1 Minimal requirements for land-surface models to mechanistically simulate TRW

The conceptual linear aggregate model of tree growth (Cook and Kairiukstis, 1990) considers that the observed TRW at year t (in mm) consists of five additive growth contributions (Fig. 2, left column) and as such provides a framework for simulating tree-ring widths with (semi)mechanistic model approaches (Fig. 2):

(i)  Size-dependent growth is the dominant signal in raw tree-ring measurements (Cook et al., 1995). Conceptually, an almost constant volume of wood due to a more or less constant primary production (Hirata et al., 2007) being added to the trunks year after year (Nash, 2011). The annual diameter increment of the trees decreases as the trunk grows wider because a given wood volume has to be distributed over an increasing surface area as both the circumference and height of the stem

are increasing. In reality, however, self-thinning tends to reduce stand density and competition for resources. The trees left can thus increase their crown volume and their primary production (Oliver and Larson, 1996) which largely compensates for the size-dependent decrease in TRW and contributes to the observed almost constant TRW of tall trees. Several of the common allocation schemes used in LSMs account for size-dependent growth and stand self-thinning (Franklin et al., 2012; Wolf et al., 2011).

(ii)      Climate-dependent growth reflects the sensitivity of tree growth to radiation, temperature, phenology, and water availability (Fritts, 2012) and is accounted for in LSMs, as it represents the core purpose of this type of models. LSMs often rely on the Farquhar model for the radiation and temperature dependency of photosynthesis (Farquhar, 1989), the McCree - de Wit - Penning de Vries - Thornley approach for the temperature dependence of respiration (Amthor, 2000). They account for a decoupling of photosynthesis and growth by the use of a labile carbon pool (Friend et al., 2019; Naudts et al., 2015; Zaehle and Friend, 2010). Plant water availability is represented through either simple transfer functions or more recently by accounting for the hydraulic architecture of the simulated trees (Bonan et al., 2014; Naudts et al., 2015).

(iii)      Endogenous disturbances refer to within-stand resource competition and are being increasingly simulated in LSMs albeit often by empirical approaches (Haverd et al., 2013; Moorcroft et al., 2001; Naudts et al., 2015). From a benchmarking point of view, simulating individuals of different size or cohorts within a single forest is essential to reproduce the sampling biases present in the ITRDB (see Section 2.2 and 2.3 below).

(iv)      Chronic exogenous disturbances such as increasing atmospheric $CO_2$ concentration (LaMarche et al., 1984) and N-deposition (Magnani et al., 2007) are well-developed as they are among the main purposes of using LSMs. The effect of $CO_2$ fertilization on photosynthesis is accounted for in the photosynthetic submodel whereas nitrogen dynamics are accounted for through static or dynamic stoichiometric approaches (Vuichard et al., 2019; Zaehle and Friend, 2010). Although abrupt disturbances such as fires, pests and storms are increasingly being simulated by LSMs (Chen et al., 2018; Yue et al., 2014) and leave marks in TRW (Bowman et al., 2009; Bräuning et al., 2016), e.g., fire scars and missing rings, they are of limited use for benchmarking against TRW data. The timing of such events largely depends on the simulated diagnostics, for example, fuel wood build-up, insect population dynamics, and soil moisture, which could strongly deviate from the observed timing in decadal to century long simulation periods. Thus simulated stand demographics should be the basis for benchmarking against observations rather than secular changes such as infrequent disturbances described above.

(v) The final term in the aggregate tree-growth model constitutes all processes and interactions between processes not previously accounted for in the LSM, and will make up the model error.

This aggregate tree-growth model provides the conceptual basis for tree-ring standardization and climate signal extraction methods used in dendrochronology (Briffa and Melvin, 2011; Cook and Kairiukstis, 1990). These methods rely on the assumptions that the sampled trees capture the common growth variability of the stand (e.g., growth responses to climate variability and resource competition), and the contribution of each major driver can be statistically identified as either signal or noise.

Alternative approaches based on Liebig's Law of the Minimum (Stine, 2019) have been proposed to attribute TRW to its major drivers. In practice, observed TRW records cannot always be fully decomposed in the absence of metadata because several drivers might not leave a unique fingerprint in growth. However, size dependent growth and climate sensitivity have been observed to comprise the primary contributions of variance in the linear aggregate model (Hughes et al., 2011).

In addition to accurate process representation, LSMs will need to be driven by historical climate, atmospheric $CO_2$ concentrations and N-deposition. In general, commonly-used century-long climate reanalyses such as NCEP (Kalnay et al., 1996), 20CR (Compo et al., 2011), and CERA-20C (Laloyaux et al., 2018) are based on the assimilation of instrumental observations in climate simulations and are thus independent from climate estimates derived from tree rings or other proxy data. Nevertheless, random and systematic errors in the reanalyses increase as data availability decreases, particularly in remote areas with a 125    low density and temporal depth of meteorological stations. Given that local climate effects may have contributed to the TRW, it might be desirable to correct the bias in reanalysis with present day site-specific climate observations where they exist (Ols et al., 2018). When LSMs are forced by actual climate observations, reproducing the observed climate sensitivity in tree rings would add credibility to the land-surface simulation – if forcing, LSM and TRW models are all realistic and unbiased.

Given the above, LSMs that intend to use TRWs as a benchmark should at the minimum simulate: (1) dynamic plant phenology, 130    (2) size-dependent growth, (3) differently-sized trees within a stand, and (4) responses to chronic exogenous environmental changes (Fig. 2). Whereas responses to chronic exogenous environmental changes are the reason LSMs exist and are therefore to some extent accounted for by all current LSMs, size-dependent growth and size differentiation within a stand are at present only accounted for in a few LSMs, for example, CLM (ED) (Fisher et al., 2015), ORCHIDEE (Naudts et al., 2015), and LPJ-GUESS (Smith, 2001). The ORCHIDEE model (revision 5698) meets the aforementioned minimum requirements and 135    therefore will be used in this study.

## 2.2    Challenges of using ITRDB data as a long-term benchmark

A typical record in the ITRDB consists of TRW measurements of increment cores from tens of individual trees from the same site and species. Each record may have different starting and ending dates, and thus length (Fig. 3). If a core reaches the pith of the trunk, annual tree diameter can be reconstructed (Bakker, 2005). Even then diameter reconstruction may come with some 140    uncertainty because trunks are not perfectly round. If the core does not contain the pith, which is often the case for large trees, the lack of information about the rings near the pith adds uncertainty to the diameter and age reconstruction (Briffa and Melvin, 2011). In this case, diameter increment could still be reconstructed (by subtracting the measured TRW from the diameter of the tree) if trunk diameter at the time of sampling was known, but this metadata is rarely recorded in dendroclimatic collections and it is not stored in the ITRDB.

The predominant sampling design in the ITRDB targets the presumably oldest trees, which should give the longest time series and are therefore most useful to reconstruct the climate variability prior to instrumental records. The ITRDB is thus likely to

over-represent large trees (Brienen et al., 2012; Nehrbass-Ahles et al., 2014) relative to the population demographics at the time of sampling. This big-tree selection bias makes the ITRDB unsuitable to upscale growth of individual trees to larger spatial domains, i.e., stand, forest or the region (Babst et al., 2014a; Nehrbass-Ahles et al., 2014) but does not affect the value of the ITRDB archive for documenting individual tree growth as long as tree size and dominance effects are explicitly considered.

Tree-ring datasets often contain cores of individuals from different cohorts. Slow and fast-growing trees within the same cohort (illustrated by the grey lines in Fig. 3) is another source of bias (Melvin, 2004; Zuidema et al., 2011). Slow-growing trees tend to live longer than fast-growing trees in the same cohort (Mencuccini et al., 2005; Schulman, 1954). Owing to survivorship being biased towards slow-growing individuals (Bowman et al., 2013), TRW records are thus likely to underestimate the mean tree growth of a stand in long-passed centuries as fast-growing trees would have died off before the samples were taken (Brienen et al., 2012).

## 2.3 Solutions for the challenges and virtual trees

Despite its known biases, poorly described sampling protocols, or protocols that were not rigorously enforced, information contained in TRW records of the ITRDB can still be used for benchmarking LSMs.

Without additional data, model-data comparison cannot correct for the big-tree selection bias in the ITRDB, however, this bias can be accommodated through models that simulate multiple tree diameter classes by comparing the largest simulated diameter class with the observed ITRDB tree-ring records (illustrated by the black dotted and highest blue lines in Fig. 4 a)

Likewise, model-data comparison cannot correct biases from slow-grower survivorship, but we propose to enhance the consistency between modelled and observed TRWs by making use of site-specific virtual trees. Virtual trees are created from observations by combining data from different individuals to obtain a time series of TRW with the desired property (details on the desired properties are given below) that reflects stand-level characteristics. By definition, virtual trees are not observed as data sequences in the ITRDB observations, but rather are extracted from site level data. Since for a single site, the tree-ring observations consist of samples from multiple individual trees (individual trees are shown by grey lines in Fig. 4), constructing a single virtual tree for a given site facilitates data-model comparisons. Because virtual trees are dependent on the chosen desired property, multiple data-model comparisons are possible for each site. In section 2.4 below, we propose four different benchmarks based on the ITRDB data, which make use of three virtual tree target properties:

- **Tree-age-aligned average virtual tree:** The average virtual tree of a stand aligned by tree age is calculated as the time series for the average ring width after aligning the age of the individual trees (Fig. 4 a). Age-aligned TRWs are widely used to calculate a statistic known as the mean regional curve of the sampled stand (Briffa and Melvin, 2011). This assumes that size-related growth is the dominant driver of tree growth (see section 2.1; subsection 2.4 (i)).

– **Calendar-year-aligned average virtual tree:** The average virtual tree of a stand aligned by calendar year is calculated by ordering individual tree-ring series by calendar year (Fig. 4 b) and for each year the average observed diameter is calculated. Alignment by calendar year thus reflects the real temporal evolution of the stand. This virtual tree can be used to better cope with a difference in simulated and observed forest structure by compiling a representative and comparable tree with the simulated tree (see subsection 2.4 (ii)).

– **Tree-age-aligned largest virtual tree:** The largest virtual tree of a stand is calculated after aligning individual trees by their age (Fig. 4 c). The recommendation to remove the age trend from tree-ring records (Cook et al., 1995) confirms the assumption underlying the alignment by age, i.e., that size-dependent age exceeds the growth trends due to long-term environmental changes. Subsequently, the age-aligned TRWs can be used to compile a virtual fast-growing tree that has the maximum observed diameter of all trees for a given tree age (note the difference in the weight of dark grey lines to virtual trees in Fig.4 b and c). The virtual fast-growing tree thus gives a better idea of the true mean tree growth in old stands. (see subsection 2.4 (iii)).

The proposed data-model comparison largely relies on the concept of virtual trees, since these virtual trees can account better for known sampling biases of the ITRDB datasets and different aspects of TRW, and facilitate comparing simulations and observations at the stand level. The proposed definitions and uses of virtual trees which were partly customized to ORCHIDEE r5698, are evaluated in section 4. Except for LSMs with an individual tree-based stand definition (Sato et al., 2007), benchmarking other models against ITRDB data will also have to consider the use of virtual trees and may have to adjust the proposed definitions to the peculiarities of the LSM under evaluation.

## 2.4 Benchmarks for comparing observed and simulated tree-ring widths

If a LSM explicitly accounts for the main contributors to TRW, i.e., size effects and climate sensitivity (Hughes et al., 2011), meaningful benchmarking against specific aspects of the observations becomes feasible in spite of the aforementioned biases in the ITRDB. Our technical framework considers four complementary aspects of the observations: (i) the size-related growth trend in tree-rings; (ii) diameter increment of mature trees; (iii) diameter increment of young trees; and (iv) extreme growth events. Each of these aspects formed the basis of a benchmark:

(i)     **Size-related diameter growth.** The size-related growth trend in diameter increment can be assessed by calculating the average virtual tree for a stand aligned by tree age (examples shown in Fig. 4a and Fig. 5 a and b) and subtracting this virtual tree from the simulated TRWs of the largest diameter class (Fig. 5 c). Subsequently, linear regression is used to quantify the temporal trend in the residuals (examples shown in Fig. 5d). If the simulations and observations have similar size-related trends, the temporal trend in the residuals will be close to zero. Furthermore, the root-mean-square error (RMSE) between the simulations and observations is calculated and normalized by the length of time series used to calculate the difference in

observed and simulated growth trends. A skilled model is expected to simultaneously show no trend in the residuals and a low RMSE.

(ii) **Diameter increment of mature trees.** In LSMs that account for within-stand competition, larger trees will consistently grow faster than smaller trees due to the way competition is formalized (Bellassen et al., 2010; Haverd et al., 2013). In reality, growing conditions can suddenly become favorable for trees that have previously been suppressed, resulting in fluctuating growth rates (see dark-grey lines in Fig. 4b). This discrepancy between simulated and observed competition can be accounted for in the benchmark by using the observations to compile a virtual tree of the stand aligned by calendar year, taking the average tree diameter of all samples to construct the virtual tree (Fig. 4b and Fig. 6 a and b). Following the big-tree selection bias (Section 2.2) it can be assumed that the observed trees are representative of the biggest trees from a given site. Hence, the virtual tree can be compared with the largest diameter class from the model. The survivorship bias of slow-growing individuals has the strongest impact when assessing TRW in century-old trees (Section 2.2). When analyzing recent decades both the quick and slowly growing trees are still alive and could have been sampled, therefore, the first five decades were excluded for the better comparison (Fig. 6 c and d). The 50-year threshold in this study is somewhat arbitrary but reflects the observation that in most of the selected time series, the fastest changes in tree growth occur in the first few decades. When benchmarking against other TRW data, this threshold could be adjusted to better fit the observed growth dynamics for other tree species and/or other regions. The RMSE and trend of the residuals between the virtual tree and the largest diameter class simulated are calculated (Fig. 6 d-f). A skilled model is expected to simultaneously show no trend in the residuals and a low RMSE.

(iii) **Diameter increment of young trees.** The diameter increment of young trees can be assessed by calculating the largest virtual tree of the stand. The maximum age of a virtual tree equals the shortest observed individual TRW record for the stand, as it represents the age intersection between the TRW records for all individuals in the stand. The largest virtual tree is clearly biased towards higher observed diameters, compensating for the loss of observed high diameters in-field sampling due to the fact that the old fast-growing trees died well before sampling took place (Fig. 4 c and Fig. 7 a and b). The first three decades of growth of the virtual tree are then compared to the simulated growth of the largest diameter class (Fig. 7 c and d) by calculating the RMSE and trend of the residuals (Fig. 7 d-f). As for the previous benchmark, the threshold is somewhat arbitrary and was set to focus the analysis on the first decades in which diameter growth is generally faster compared to later decades. Note that the thresholds for young and mature trees are separated by 20 years during which the observations are not considered in either benchmark. When needed, these thresholds could be adjusted to better match local tree growth, but it is recommended to keep the separation between the thresholds to account for the transition in diameter growth from young to mature trees. A skilled model is expected to simultaneously show no trend in the residuals and a low RMSE. By using different approaches to evaluate the growth of young (this benchmark) and mature trees (the previous benchmark) the comparison accounts for the observation that the drivers of ring growth change as the trees grow taller (Cook, 1985).

(iv) **Extreme growth events.** For this benchmark, extreme growth is here defined coarsely as the first and last quartiles in TRW ordered by calendar year. For the purpose of estimating extremes, we seek the average virtual trees and LSM sim-

ulations to most accurately and precisely characterize the interannual variability. In the present study, use of the 1951-2000 interval produces LSM simulations forced by climate and nutrient loading estimates best constrained by dense, recent climate observations (see Section 3), and therefore minimizing the contribution of forcing uncertainty to LSM simulation uncertainty; in the TRW observations, it also leverages non-juvenile tree-ring series of more than 50 years which do not require detrending and are expected to most reflect the climate sensitivity. Subsequently, for each site, individual tree records are averaged to obtain a single time series (Fig. 8a). Model skill for estimating the distribution of growth arising from climate variability is evaluated by comparing the observed and simulated 25th and 75th percentiles of TRWs for the largest diameter class, which is the diameter class showing the strongest climate sensitivity (Fig. 8 e and f). Additionally, model skill for reproducing the timing of individual extreme growth events is evaluated by comparing simulated to observed virtual standardized TRW for the exact years during which extreme growth was observed (Rammig et al., 2015, Fig.7 a-d). For both the amplitude and timing of growth extremes, the similarity between simulations and observations was calculated as the RMSE of the distance from the 1:1 line (Fig. 8 c-f). A skilled model is expected to simultaneously show low RMSE for both the amplitude and timing of extreme years.

## 3    Materials and Methods

### 3.1    The land-surface model ORCHIDEE

ORCHIDEE (Ducoudré et al., 1993; Krinner et al., 2005) is the land-surface model of the IPSL (Institute Pierre Simon Laplace) Earth system model (Dufrêne et al., 2005). Hence, by design, it can be coupled to an atmospheric general global circulation model or become a component in a fully coupled Earth system model. In a coupled setup, the atmospheric conditions affect the land-surface and the land-surface, in turn, affects the atmospheric conditions. However, when a study focuses on changes in the land-surface rather than on the interactions with climate, it can also be run as a stand-alone land-surface model. In both configurations the model receives as input atmospheric conditions such as precipitation, air temperature, air humidity, winds, incoming solar radiation, and $CO_2$; this combination of inputs is known as the climate forcing. Both configurations can cover any area ranging from global to regional domains and even down to a single grid point for the stand-alone case.

Although ORCHIDEE does not enforce a spatial or temporal resolution, the model does use a predefined spatial grid and equidistant time steps. The spatial resolution is an implicit user setting that is determined by the resolution of the climate forcing. Although the temporal resolution is not fixed, the processes were formalized at given time steps: half-hourly (i.e. photosynthesis and energy budget), daily (i.e. net primary production), and annually (i.e. vegetation dynamics). Hence, meaningful simulations have a temporal resolution between 1 minute and 1 hour for the energy balance, water balance, and photosynthesis calculations.

ORCHIDEE builds on the concept of meta-classes to describe vegetation distribution. By default, it distinguishes 13 meta-classes (one for bare soil, eight for forests, two for grasslands, and two for croplands). Each meta-class can be subdivided

into an unlimited number of plant functional types (PFTs). When simulations make use of species-specific parameters and age classes, several PFTs belonging to a single meta-class will be defined. Biogeochemical and biophysical variables are calculated for each PFT or groups of PFTs (e.g. all tree PFTs in a pixel drawn from the same description of soil hydrology, known as a soil water column).

ORCHIDEE is not an individual-based model but instead it currently represents forest stand complexity and stand dynamics

with diameter and age classes. Each class contains a number of individuals that represent the mean state of the class. Therefore, each diameter class contains a single modelled tree that is replicated multiple times and distributed at random throughout the PFT area. At the start of a simulation, each PFT contains a user-defined number of stem diameter classes. This number is held constant throughout the simulation, whereas the diameter boundaries of the classes are adjusted to accommodate for temporal evolution in the stand structure. By using flexible class boundaries with a fixed number of diameter classes, different forest

structures can be simulated. An even-aged forest, for example, is simulated with a small diameter range between the smallest and largest classes. All classes will then effectively belong to the same stratum. An uneven-aged forest is simulated by applying a large range between the diameter classes. Different diameter classes will therefore effectively represent different strata. The limitations of this approach become apparent when the TRW data and simulation are compared by calendar year as the model does not track individual trees. Although the dimensions of each model tree itself are well-defined, the amount of radiation it

receives (and therefore the amount of carbon produced) is determined by the statistical distribution of all model trees in that grid cell.

Vegetation structure is then used for the calculation of the biophysical and biogeochemical processes of the model such as photosynthesis, plant hydraulic stress, and radiative transfer model. The r5698 version of ORCHIDEE (Text S1), used in this study, combines the dynamic nitrogen cycle of ORCHIDEE r4999 (Vuichard et al., 2019; Zaehle and Friend, 2010) and the

290 explicit canopy representation of ORCHIDEE r4262 (Chen et al., 2016; Naudts et al., 2015; Ryder et al., 2016). It simulates carbon, water, energy, and nitrogen fluxes, allows for size-dependent allocation across three diameter classes within a forest stand, and has been parameterized and tested for the simulation of TRW series via radial tree growth estimation. A detailed overview of earlier developments (Krinner et al., 2005; Naudts et al., 2015; Vuichard et al., 2019) that resulted in the emerging capability of ORCHIDEE r5698 to match the aggregate tree growth model (Fig. 2) is given in the supplementary material (Text

S1).

### 3.2 Simulations: forcing and model parameterization

In this study, offline ORCHIDEE simulations for the 20th century are forced with the merged and homogenized gridded CRU-NCEP climate dataset (Viovy, 2016); the gridded nitrogen deposition product from the Chemistry-Climate Model Initiative

(CCMI) (Eyring et al., 2013); a gridded nitrogen fertilization product for $N_2O$ (Lu and Tian, 2017); an observation-based time
series of global atmospheric $CO_2$ concentrations (Keeling et al., 1996); forest management followed the reported management
status for each of 11 sites simulated (Table S3) for comparison, via the virtual tree benchmarks, with observations (Section
3.3). Simulations were started from a 300-year long spinup, which was required for equilibrium with respect to the slow carbon
and nitrogen pools in the soil. The spinup was concluded with a forest clear cut, such that the start year and the length of each
simulation matched the observed stand age for the validation dataset (Section 3.3).

During model development, two global (number of diameter classes and ratio for number of trees per each diameter class;
see 3.1) and six PFT-specific parameters ($f_{power}$, $f_\sigma$, $k_{\alpha\_r}$, $k_{\alpha\_s}$, $k_{\beta\_r}$, and $k_{\beta\_s}$; Table S1) were manually adjusted to jointly
reproduce the TRW data of 11 ITRDB sites: aust112 (Strumia, 2005), cana106 (Archambault and Bergeron, 2002), chin037
(2k Consortium, 2013), finl055 (Melvin, 2005), fran4 (Tessier, 1996), id007 (Briffa and Schweingruber, 2002), japa011 (Davi
et al., 2011), mo009 (Bell et al., 2002), nepa003 (Krusic and Cook, 2005), spai055 (Briongos and Cerro-Barja, 2007), and
turk027 (Griggs et al., 2006). Sites belonging to the same PFT were thus simulated by making use of a single PFT-specific
parameter set. In other words, no site-specific parameters were applied. All other parameters were set to default values. As
configured, the model distinguished five diameter classes for simulated trees. The smallest and largest diameter classes each
contained 15% of the total number of simulated trees. The three intermediate classes contained 21%, 27% and 21% of simulated
trees respectively. Agreement between simulated and observed TRW was assessed visually.

## 3.3   Verification

The European biomass network contains TRW samples from "fixed-plot sampling". The database was established within
multiple research projects and made publicly available through the EU Horizon-2020 project BACI (BACI, 2020). It archives
at present 48 datasets covering temperate and semi boreal climates (Fig. S2) and being collected from a variety of research
efforts in Eurasia (Klesse et al., 2018). All trees larger than 5.6 cm in diameter at breast height had to be sampled in a 10 to 40
m radius plot, depending on stand density, to be archived in the BACI database (Babst et al., 2014b). The European biomass
network is, therefore, considered to be free from the big-tree selection bias that has plagued the ITRDB, although other known
biases (e.g. slow-grower survivorship bias) may still be present. The records from the European biomass network are thus
suited to evaluate the validity of using virtual trees constructed from ITRDB records to cope with the aforementioned sampling
biases.

We selected sites from the European biomass network based on the following criteria: (1) the site had to be dominated by
a single species for enhanced compatibility with ORCHIDEE, which is monospecific by design; and (2) stand age should
exceed 50 years as a requirement to apply all four proposed benchmarks (Section 2.4). The benchmarks were applied to a
common evergreen and a common deciduous species. Hence, within the filtered sites, only sites dominated by Picea Abies or
Fagus Sylvatica were retained, resulting in 12 sites out of the total of 48 sites. CIM, a site dominated by Fagus Sylvatica, was

removed from the selection (decreasing the final number of sites to 11) because only one tree out of 61 trees was aged over 100 years, resulting in a diameter distribution that is not at all compatible with the default diameter distribution of the model. The details of the selected sites are in Table S3.

The European biomass network data were additionally used to verify whether the big-tree selection bias that is present in the ITRDB data invalidates its use for benchmarking LSMs. The verification checked whether changes in parameter values or
model process representation that would be required to make the model output better match the ITRDB data, would also result in a better match between the model output and the all tree data from the European biomass network. If this would be the case, benchmarking LSMs against ITRDB data would result in model changes that would enhance the model's capability to simulate tree growth, thus justifying the conclusion that despite the known biases, ITRDB data can be used for LSM benchmarking.

The verification, therefore, used the data from the European biomass network in two different ways: 1) all trees in the European
biomass network dataset were used (hereafter called "all-tree data") to calculate the four proposed benchmarks at the site level. The results of these benchmarks were used as the reference in the verification, and 2) only big trees were sub-sampled from the data (hereafter called "big-tree data") and all four benchmarks were calculated against this sub-sample of data. Big trees were defined as the top 15% of the trees based on their diameter, and the 15% threshold was taken to match the diameter distribution in ORCHIDEE, where by definition the largest diameter class contains 15% of the trees.

The "big tree" verification required three additional steps (Fig. 9): 1) The simulated TRW from the largest diameter class were transformed by modifiers to minimize the two metrics of each benchmark (Section 2.4; Table S2). The different benchmarks may use different metrics, i.e., the RMSE and slope of the residuals were used as the metrics for benchmarking size-related growth trend, growth of mature trees, and growth of young trees, whereas extreme growth and TRW amplitude were used as the metrics for benchmarking extreme growths (Table S2). 2) The same modifiers were then applied to all simulated diameter
classes, and all four benchmarks. Hence, for each benchmark its two metrics were calculated using all-tree data, and 3) the actual verification tested whether for a given metric and a given benchmark the modifier improved simulations for the big-tree sample and for the all-tree data. Improvement of a specific metric of a benchmark was quantified by subtracting the original value for that metric from its modified value for all-tree data. A negative value thus indicated an improvement. If all three conditions were satisfied, the benchmarks of the big-tree and all-tree data were said to be consistent, implying that
using this benchmark in combination with the ITRDB data would reveal the same model shortcomings as benchmarking ORCHIDEE against TRW data from all-tree networks. Across the 11 sites and for each of the four proposed benchmarks and both benchmark metrics, sites where the test improved for both datasets were counted to estimate the confidence in using ITRDB in benchmarking LSMs.

Since this study aims to propose benchmarks making use of the ITRDB study rather than improving the ORCHIDEE model,
the modifiers were applied to the model output directly. This approach has the advantage of remaining conceptual, avoiding the need to optimize specific model parameters or rewrite or add processes in the model code. Different modifiers were used to

accommodate the differences between the metrics: 1) the RMSE or amplitude of a benchmark was minimized by multiplying the simulated TRW with a modifier (Fig. 9), 2) the slope of the residuals of a benchmark was minimized by subtracting a trend-modifier from the simulated growth trend and 3) the years of the simulated TRW were rearranged such that they matched the ranked order of observed extreme TRWs.

## 4 Results

Verification of ORCHIDEE-based TRW simulations was applied at the 11 sites selected from European biomass network (Section 3.3), via estimation of the four benchmarks (Section 2.4) from simulations and observations, their evaluation via the two skill metrics per benchmark, for a total of 88 test case comparisons (Table 1, Fig. S3). We describe the results of big-tree bias estimation and modifier estimation in general, and then for each of the individual benchmarks in the remainder of this section.

### 4.1 Big-tree bias estimation

Despite its simplicity, the use of modifiers was found to be robust as it improved all metrics of the four proposed benchmarks at each of the eleven sites when verifying against the big-tree data. Applying the same modifiers to the all-tree data improved the match between the simulations and observations in 72% of the test cases (63 out of the 88 test cases; Table 1). We note that this overall result hides large differences between tree species. The verification appeared to be more successful for beech with an overall confidence level of 84% (27 out of 32 test cases) compared to spruce with a 64% (36 out of 56 test cases) confidence level.

### 4.2 Verification by benchmark

when benchmarking size-related growth, conclusions would be similar in 16 out of 22 cases (73%), regardless of whether OR-CHIDEE is benchmarked against the big-tree data rather than the all-tree data. Some sites such as DEO and DVN showed positive differences close to zero, suggesting that simulations with ORCHIDEE r5698 matched the observed size-related growth trend reasonably well, leaving limited room for improvements. One site (SCH) showed a positive difference because it contained two slowly growing trees which lived roughly 40 years longer than the rest of trees but whose diameter was too small to be contained in the big-tree sample (Fig. S4). Except for this site, the other sites showed marginal inconsistencies or showed improved simulated output against the two datasets. The size-related trend in tree growth, thus, can be derived from either the big-tree or the all-tree data.

For the mature trees benchmark, big-tree data can be used with 68% (15 out of the 22) confidence for benchmarking against LSMs. At 5 out of 11 sites, the all-tree data and the big-tree data yield different results. Two sites (HD2 and TIC) for which for both metrics inconsistencies between the big-tree and all-tree data were observed, have 36% to 44% of small trees in their size distribution, compared to an average 28% at the other nine sites. The proportion of small trees in the observation was estimated by counting trees in the smallest bin when trees are divided into five size classes similar to the model. The site labelled as ZOF has a bimodal size distribution with the biggest number of trees in the 1st and 4th diameter class (35% and 32% respectively). The default size distribution in ORCHIDEE has 15% of its trees in the smallest-sized class, and 21% in the 4th sized-class. At two other sites, DEO and SOB, the growth rate for big trees was higher in the observations (0.95 and 0.50 for the slope of residuals) since the difference in big trees and small trees are bigger in the observations (Fig. S5 b), despite the average simulation matched well with the average diameter trend as shown by the calculated slope of residuals: 0.08 and 0.09 (Fig. S5 a). These results suggest that the mature trees benchmark is sensitive to the stand structure.

With 50% (11 out of 22) confidence in using the big-tree data in benchmarking LSMs, the young trees benchmark appears to be the most demanding in terms of its data. At sites DEO, HD2, and SOB, inconsistencies between benchmarking the big-tree data and the all-tree data stemmed from: (1) similarity between simulations and observations, with RMSE around 10 mm, and (2) the difference between big trees and small trees growth being larger in the observations (Fig. S6). The site labelled as SCH contained two extremely fast-growing young trees resulting in a very fast-growing virtual tree in the optimized model output (Fig. S7). For SOR the difficulties may have come from the model itself, more specifically from difficulties in simulating the carbon allocation (Fig. S8). These results suggest that a variety of issues decreases the confidence in using big-tree data for young tree benchmarking.

For the extreme growth benchmark, big-tree data can be used with 95% (21 out of the 22) confidence for benchmarking LSMs. The observed consistency between benchmarking the big-tree data and the all-tree data suggests that extreme growth happens in the same years, irrespective of which dataset is being used. The site (DVN) showed the smallest RMSE for amplitude when it is calculated with the all-tree data (0.02) but the site has the biggest ratio of big-trees to all-trees for amplitudes compared to simulation (1.30, Fig. S9). In other words, if the simulation is adjusted to the big trees in the observation, since the difference between sub-sampled big-tree and all-tree is larger in the observations, the average simulation becomes bigger than the average observation, as shown in Fig. S5 and S6. This result suggests that the extreme growth benchmark is the least demanding benchmark in terms of sampling design.

## 5    Discussion

### 5.1    LSM verification: beyond tree-ring width

Until now, verification of LSMs against tree ring width records has relied primarily on interannual variation in simulated net primary productivity as a proxy for site-level TRW chronologies (Klesse et al., 2018; Kolus et al., 2019; Rammig et al., 2015; Zhang et al., 2018). Although such an indirect approach is appropriate, to a certain extent, for validating the capability of LSMs to simulate interannual variability, for studying patterns and mechanism of change over longer timescales, the observations will need to be detrended to remove the size-related growth signal, adding considerable uncertainty to the verification process (Bunde et al., 2013; Cedro, 2016; Nicklen et al., 2019; Stine, 2019). Recognizing Cook's conceptual model of aggregate tree growth (Fig. 2), we propose to move beyond the net primary production proxy by explicitly simulating and validating stem radial growth demographics. By doing so, we potentially enrich the verification by including the effects of potentially confounding factors such as forest structure, age and size trends (Alexander et al., 2018; Nickless et al., 2011; Jiang et al., 2018), phenology (Shen et al., 2020) and sampling biases (Babst et al., 2014a), in addition to climate and environmental forcing (Klesse et al., 2018; Zuidema et al., 2020; Li et al., 2014; Rollinson et al., 2017).

Targeting both size-structured and age-structured information in observations and simulations (Fig. 3), we have proposed the use of four verification benchmarks created from observations and potentially simulated by LSMs, each of them defined by two complementary metrics (Fig. 2; Table S2).

(i)      The size-trend benchmark targets the long-term trend in TRW. This trend contains information about ontogenetic growth during establishment and endogenous competition from canopy closure (Cook and Kairiukstis, 1990). Although this trend is removed in many dendrochronological studies to amplify the climate signal contained in TRW (Briffa and Melvin, 2011), we suggest testing the skill of the model in reproducing it because it is important to constrain biomass production. Benchmarking a suitable LSM against observed size-related trends in TRW may help to develop, evaluate or parameterize allometric relationships and changes in simulated stand density (Fig. 4a).

(ii)      The mature trees benchmark tests the capability of the model in simulating annual growth of a mature forest. Since this benchmark aligns the observations by calendar year (Fig. 4b), it may reflect the effects of long-term environmental changes, if there were any and if the observational record is long enough to express them (Hess et al., 2018; Panthi et al., 2020). As a skilled LSM is expected to reproduce plant responses to long-term environmental change, this benchmark could be used to develop, evaluate and parameterize the processes that simulate endogenous disturbances and plant responses to, e.g., increasing atmospheric $CO_2$ concentrations, atmospheric N deposition and warming.

(iii)      Tree growth during stand establishment can be tested with the young trees benchmark. The growth of establishing trees differs from that of mature canopy trees, and this difference has been accounted for by using separate benchmarks for

young and mature stands (Fig. 4c). This benchmark could be used to develop, evaluate or parameterize allometric growth of young trees as well as tree mortality prior to canopy closure.

(iv)       The extreme growth benchmark tests the occurrence and range of extreme growth events. Previously, interannual variability in TRW has been used to evaluate the climate sensitivity of LSMs. Inter-annual variability has a limitation because we cannot expect the model to simulate either the timing of endogenous or exogenous disturbances, such as fire, pest and

disease outbreak, and death of big trees leading to sudden growth releases in adjacent trees. By forming a benchmark from the extrema of the empirical distribution of incremental growth in mature trees (e.g. as evident from Fig. 5a; see also Figs. 9, S8, and S9), we create a direct comparison with the simulated demographics of trees, as observed over a contemporaneous time interval. This benchmark could be used to develop, evaluate or parameterize the plant water stress and the temperature dependency of plant growth in the model.

The metrics of the first three benchmarks are RSME and slope (Figs. 5, 6, and 7). RMSE examines if the model reproduces the absolute values of TRWs. However, even though a model might reproduce well the value of TRWs, it is still expected to simulate the long-term trend in TRW that comes from climate changes or endogenous competition. This latter aspect is quantified by the slope-metric. For the large-scale models such as a LSM and for sites with little high-quality site information, correctly simulating growth trends should be prioritized over matching the endpoints in tree diameter. Since the benchmark for

extreme growth events was not intended to test the capability of LSMs to simulate growth trends, the slope of residuals was not included. As a skilled model is expected to simulate not only the timing of extreme growth but also the magnitude of it, the metrics for this benchmark therefore were designed to both be evaluated using RMSE (Fig. 9).

## 5.2    Toward LSM verification using the ITRDB

Regardless of the approach to LSM verification, the largest publicly available archive of tree-ring records, the ITRDB, is prone

to sampling biases (Klesse et al., 2018; Zhao et al., 2019). Although it may be difficult to correct the data for these biases, our benchmarks present two solutions for comparing LSM output to ITRDB observations of raw ring width. Simulating a size-structured population of trees enables comparing the observations relative to a benchmark for a tall simulated tree, which compensates for the tendency of dendroclimatic sampling to select the oldest trees in a stand, which may turn out to be the larger trees. Although the ITRDB does not contain the site metadata that would be required to make this comparison exact,

i.e., the diameter and true age distribution of the sampled stand, use of the tall tree benchmark protects against comparing the observed mean of a biased sample to the observed mean in unbiased simulation demographics. The second solution relies on the observation that the variation due to size-related growth by far exceeds the variation due to environmental changes and helps to constrain the survivor bias, which is derived from the growth of young fast-growing trees that died a long time ago and are therefore absent from records made from present day sampling of old growth forests (Brienen et al., 2017). The

benchmarks proposed here provide a tool to start using ITRDB TRWs as a much-needed large-scale constraint on the maximum tree diameter and annual growth for the transition from pre-industrial to present-day environmental conditions.

Our verification approach estimated the level of confidence for each benchmark from the fraction of cases for which scaling simulations to observational benchmarks for big-tree data would result in improving the model performance for observational benchmarks for all-tree data. In other words, big-tree-biased verification data should not degrade model performance relative 480 to all-tree verification data, and such tests were performed using the European biomass network dataset (BACI, 2020). The results, then, might inform the use of the ITRDB, which is suspected to contain large-tree bias, for large-scale verification of LSMs on decadal to centennial timescales (Fig. S1b).

Verification results (Table 1) show that if the output of ORCHIDEE is benchmarked against data with imposed big-tree bias, there is 70% confidence that the benchmark will produce similar conclusions as reached from use of all-tree data (Table 1, 485 columns 1-2). This level of confidence is perhaps sufficient to support benchmarking a LSM against tens to hundreds of ITRDB sites in aggregate. The same level of confidence is likely too low to benchmark a LSM, or any ecosystem model, against a single ITRDB data series, as there is a 30% chance that parameter tuning or model improvements following the benchmarking will not verifiably improve the model. Given the limited spatial extent, species content, and environmental range of the European biomass network used in this study, the levels of confidence represent temperate and hemi-boreal forests, a subset of the ITRDB 490 range and the actual distribution of forests (Fig. S2). The validity of the use of the ITRDB data from boreal and tropical forests will need to be verified as suitable data become available.

Across all species, benchmarking against extreme events, mature trees, and the size-related growth appeared to be the least sensitive to big-tree bias (Table 1; bolded 63/88 cases). Benchmarking against young trees will benefit from using data free from the big-tree selection bias (Table 1, compare bolded values reported in columns 1-4 and 7-8 to those in columns 5-495 6). This finding suggests limited utility of ITRDB data to verify simulations using the young tree benchmarks. Because the true diameter distribution is not contained in the ITRDB, it is neither possible to select only ITRDB sites for which the actual diameter distribution matches ORCHIDEE's distribution, nor to adjust the diameter distribution in ORCHIDEE to the observed distribution (Figs. S4 to S7). Matching observed and simulated distributions appears to be essential when benchmarking the growth of young trees. The same finding suggests, however, that forest inventory data for which the diameter distribution is 500 known but only a few big trees were cored would be a reliable data source for benchmarking LSM. The higher fraction of modifier-improved ORCHIDEE simulations for beech (87%) relative to spruce (64%) (Table 1, compare bolded values in rows 1-7 to bolded values in rows 8-11) suggest that the validity of the assumptions underlying the use of ITRDB data partly depends on tree species. Unfortunately, in this study, the variety in species in the was too limited to generalize this result in terms of plant functional types.

Even when benchmarks calculated from observations and simulations are not in agreement, they may nevertheless be used to identify ways in which to improve the observational database or the simulation model. Considering the comparison of observed

to simulated interquartile range (Fig. S9), we see that the GIU site is amongst the poorest simulated sites. However, we found that the simulation could be improved for both metrics of all four benchmarks through the use of modifiers (Table 1, third row), despite the difference in simulated and observed stand structure (Fig. S10). Yet we also found that inconsistencies between big-tree data and all-tree based benchmarks may appear even when the simulated mean TRWs approach observed means (Figs. S4 to S7). This suggests that: 1) ITRDB data can be used as a first approximation to benchmark the growth of young and mature trees in LSMs, 2) as the model improves, the need for unbiased datasets will increase, as biases in observed stand structure and growth rates could hamper the use of young and mature tree benchmarks in particular.

## 5.3  Outlook

Tree-ring records of incremental growth, suitably described in terms of benchmarks that more fully describe the information richness of these datasets, might complement well-established but short-term benchmarks for LSMs (Randerson et al., 2009), such as forest inventory data (Bellassen et al., 2010; Naudts et al., 2015), eddy covariance measurements (Blyth et al., 2010; Williams et al., 2009), Free Air $CO_2$ Enrichment experiments (De Kauwe et al., 2013) and satellite observations of vegetation activity (Chen et al., 2011; Demarty et al., 2007). The novel benchmarks proposed here may also provide new targets for evaluating LSMs' performance, as the metrics could be used in the objective function of any data assimilation technique (Peylin et al., 2016) to rigorously account for the information contained in TRW datasets. The value of tree-ring records for LSM verification might be further enhanced by: (i) developing new, unbiased networks, such as the European biomass network, to both complement and identify biases in the ITRDB, (ii) adding their stable isotope ratios to verification benchmarks that may be simulated by isotope-enabled LSMs (Levesque et al., 2019; Barichivich et al., 2020) and (iii) combining their use with high-frequency but short-term eddy covariance measurements (Pappas et al., 2020; Teets et al., 2018), experimental data from plant growth under pre-industrial $CO_2$ concentrations (Temme et al., 2015), and proxies of atmospheric composition (Campbell et al., 2017).

## 6  Conclusion

We have proposed and evaluated the use of four benchmarks and two metrics that leverage observed demographics to provide more nuanced verification targets for LSMs that simulate both demographic responses and their environmental forcing on decadal to centennial timescales. Using small but relatively unbiased European biomass network datasets, we identify the extent to which presumed biases in the much larger ITRDB might degrade the validation of LSMs. We find that size, mature tree and extreme growth verification benchmarks are relatively insensitive to big-tree bias, but use of young tree benchmarks for verification of LSMs may require development of new unbiased observational TRW datasets, and/or innovative use of independent verification data.

## 7 Code and data availability

In line with GMD requirements, the model code has been archived and made accessible: https://doi.org/10.14768/20200228001.1. The scripts required for reproducing the figures, the ORCHIDEE simulations and intermediate results are available at https://github.com/j-jeong/J.Jeong_GMD_2020. BACI dataset is freely available in online: http://www.baci-h202i0.eu/ but requires
registration by email.

## 8 Author contribution

Proposed benchmarks are the outcome of discussions between JJ, JB, PP, VH, and SL. JJ ran the model, analysed the output and prepared figures. FB collected the BACI data. JJ, SL and MNE wrote the manuscript, all authors contributed to revising and editing the different versions of the manuscript.

## 9 Competing interests

The authors declare that they have no conflict of interest.

*Acknowledgements.* JJ, PP, MJM and SL were funded by the VERIFY project under the European Union's Horizon 2020 research and innovation program under grant agreement No. 776810. JB was supported by the Centre National de la Recherche Scientifique (CNRS) of France through the program "Make Our Planet Great Again". VH acknowledges support from the Earth Systems and Climate Change Hub, funded
by the Australian Government's National Environmental Science Program. MNE was supported by NSF/AGS1903626 and the University of Maryland, and acknowledges insights arising from work with the PAGES/Data Assimilation and Proxy System Modeling Working Group. F.B. acknowledges funding from the project "Inside out" (#POIR.04.04.00-00-5F85/18-00) funded by the HOMING program of the Foundation for Polish Science co-financed by the European Union under the European Regional Development Fund. Stefan Klesse co-developed the European biomass network and provided management information. SL would like to thank Antonio Lara (Universidad Austral de Chile) for
early discussions on the topic. Valérie Daux (Laboratoire des Sciences du Climat et de l'Environnement) is acknowledged for commenting on a previous version of the manuscript which improved its clarity.

**Table 1.** Verification of the benchmarks and their metrics. Each cell represents the result from a single site. The values show the difference for each metric before and after optimization. Bold cells show the cases where the optimization for the all-tree data was consistent with the optimisation result of the big-tree data.

| Benchmark | Size dependent growth | | Diameter increment of mature trees | | Diameter increment of young trees | | Extreme growth | |
|---|---|---|---|---|---|---|---|---|
| Metrics | RMSE (mm) | Slope of residuals (mm/yr) | RMSE (mm) | Slope of residuals (mm/yr) | RMSE (mm) | Slope of residuals (mm/yr) | Amplitude (mm) | Extreme growth (scaled) |
| *Picea abies* | **DEO** **(-0.005)** | DEO (0.000) | **DEO** **(-75.97)** | DEO (0.78) | DEO (8.11) | DEO (1.47) | **DEO** **(-0.04)** | **DEO** **(-0.60)** |
| | **DVN** **(-0.182)** | DVN (0.000) | **DEO** **(-161.69)** | **DVN** **(-0.69)** | **DVN** **(-11.96)** | **DVN** **(-0.42)** | DVN (0.02) | **DVN** **(-0.77)** |
| | **GIU** **(-0.600)** | **GIU** **(-0.007)** | **GIU** **(-131.25)** | **GIU** **(-0.68)** | **GIU** **(-39.96)** | **GIU** **(-1.30)** | **GIU** **(-0.32)** | **GIU** **(-0.96)** |
| | HD2 (0.009) | **HD2** **(-0.002)** | HD2 (15.60) | HD2 (0.53) | HD2 (1.95) | HD2 (0.56) | **HD2** **(-0.04)** | **HD2** **(-0.97)** |
| | SCH (0.029) | **SCH** **(-0.004)** | **SCH** **(-182.46)** | **SCH** **(-0.96)** | SCH (57.55) | SCH (5.49) | **SCH** **(-0.15)** | **SCH** **(-1.29)** |
| | **SOB** **(-0.008)** | SOB (0.001) | **SOB** **(-20.22)** | SOB (0.49) | SOB (9.32) | SOB (1.29) | **SOB** **(-0.08)** | **SOB** **(-1.54)** |
| | **TIC** **(-0.151)** | **TIC** **(-0.001)** | TIC (24.47) | TIC (1.52) | **TIC** **(-10.33)** | **TIC** **(-0.28)** | **TIC** **(-0.19)** | **TIC** **(-1.63)** |
| *Fagus sylvatica* | **CAN** **(-0.046)** | **CAN** **(-0.004)** | **CAN** **(-74.03)** | **CAN** **(-1.27)** | **CAN** **(-5.91)** | CAN (0.16) | **CAN** **(-0.07)** | **CAN** **(-1.17)** |
| | SOR (0.007) | **SOR** **(-0.005)** | **SOR** **(-116.26)** | **SOR** **(-1.62)** | SOR (2.69) | **SOR** **(-1.13)** | **SOR** **(-0.04)** | **SOR** **(-1.01)** |
| | **TER** **(-0.060)** | **TER** **(-0.000)** | **TER** **(-3.73)** | **TER** **(-0.09)** | **TER** **(-15.93)** | TER (0.25) | **TER** **(-0.07)** | **TER** **(-0.99)** |
| | **ZOF** **(-0.183)** | **ZOF** **(-0.000)** | **ZOF** **(-42.72)** | ZOF (0.02) | **ZOF** **(-11.98)** | **ZOF** **(-0.05)** | **ZOF** **(-0.17)** | **ZOF** **(-1.11)** |

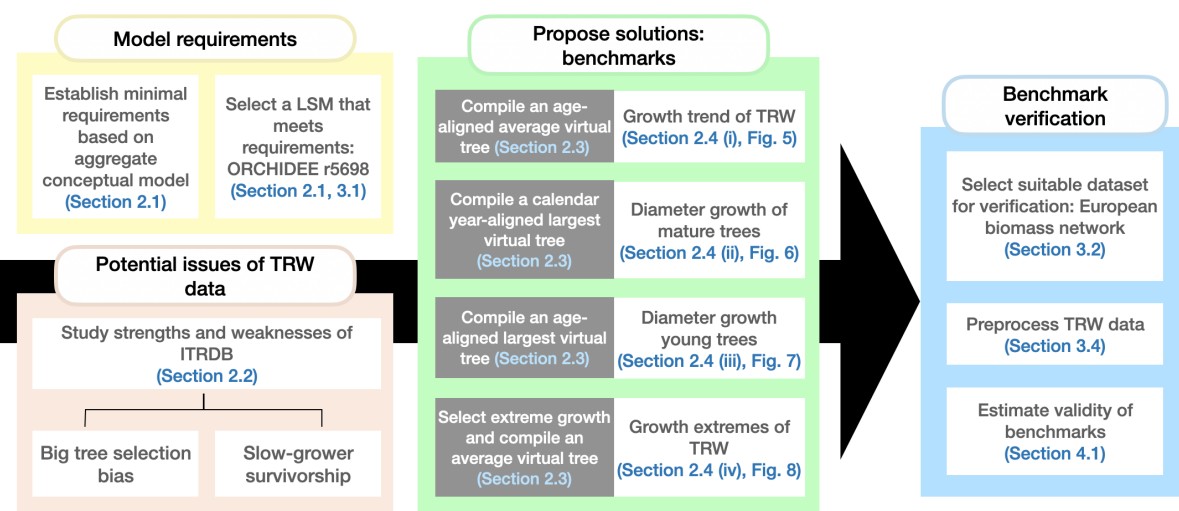

**Figure 1.** Workflow of this study and the dependencies between the different sections of the manuscript. Each colour represents a different aim of the study. The arrow shows that the outcomes of the first three aims have to be combined to verify the proposed benchmarks. In this study, "virtual tree" refers to a tree that does not exist in the data but is constructed from the data in varying ways. Section 2.3 provides detailed descriptions of the different virtual trees used.

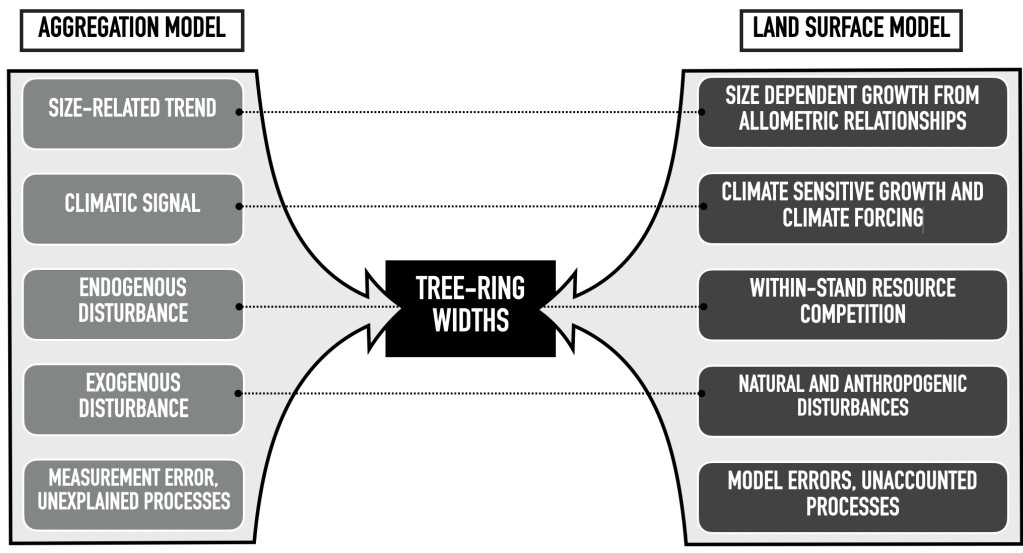

**Figure 2. Main drivers of the linear aggregate conceptual model of tree-ring growth (Cook and Kairiukstis, 1990) and the equivalent processes in land-surface models.** The dotted lines connect related components. Note that both the aggregation and the land-surface model come with errors, uncertainties and unaccounted processes which are not explicitly modelled.

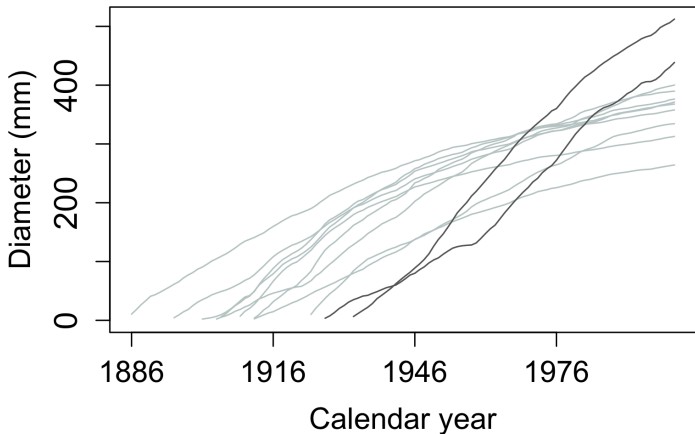

**Figure 3. Example of a typical data record in the ITRDB dataset.** Each dataset in ITRDB consists of increment cores from multiple trees (tens to several hundred, depending on the dataset) with varying ages and growth rates. In this figure, observations are from a pine forest in Germany archived as germ214 (Neuwirth et al., 2007) (Table S3). The diameter of individual trees was reconstructed by summing the annual tree-ring from the dataset. Note the presence of both fast-growing relatively young trees (dark-gray lines) and slow-growing older trees (light-gray lines).

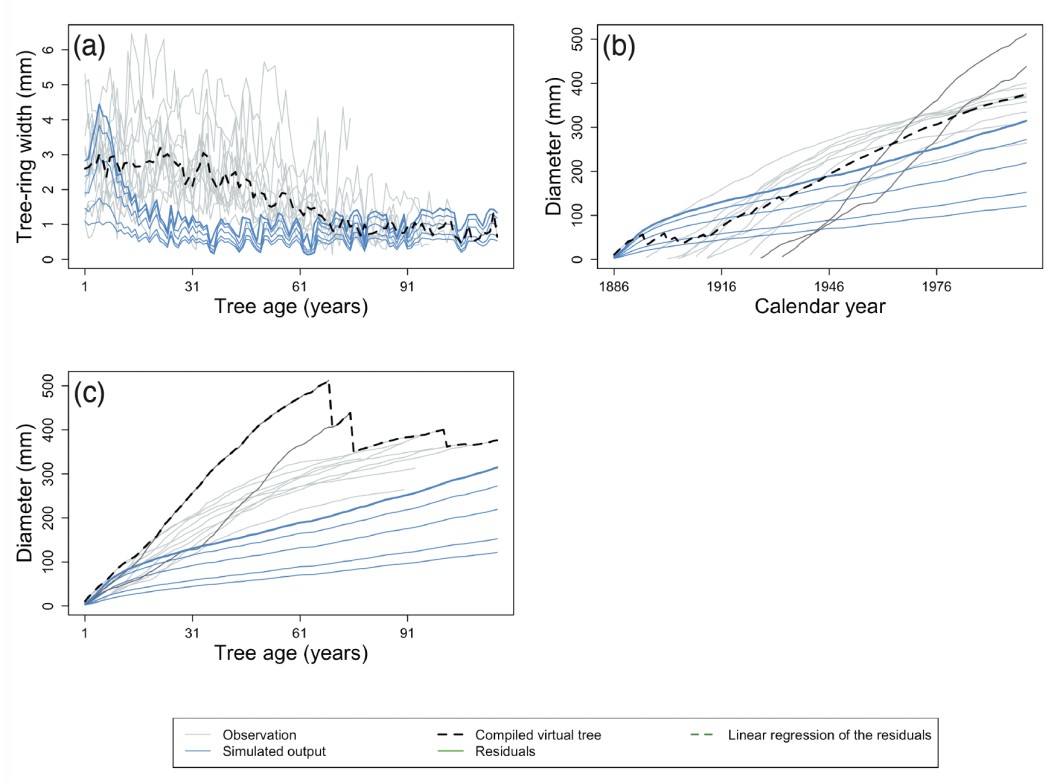

**Figure 4. Using virtual trees to account for challenges related to the use of ITRDB datasets when evaluating land-surface.** (a) Data-model comparison may overcome the big-tree selection bias by comparing only the simulated largest diameter class (bold blue line) for evaluation rather than all diameter classes (thin blue lines), with the compiled average virtual tree (black dotted line). Grey lines represent individual trees from observations. (b) The observed tree-ring records are a mixture of relatively slowly-growing trees (light-grey lines) and fast-growing trees (dark-grey lines). Fast-growing trees don't attain the same age as slowly-growing trees because they tend to die earlier. Using all trees without further consideration in the calculation of the average virtual tree (black dotted line) would lead to underestimating tree growth at the time of stand establishment resulting in a flawed comparison with the simulated tree growth (blue lines). (c) However, aligning observations by the age of individual trees before compiling largest virtual tree (black dotted line) results in a very different virtual tree compared to Fig 3b. The largest virtual tree better reconstructs tree growth during stand establishment, facilitating data-model comparison. Note the change in the label of the X-axis between panels (b) and (c). Observations taken from a pine forest in Germany archived as germ214 (Neuwirth et al., 2007) (Table S3). Simulation were run with ORCHIDEE r5698.

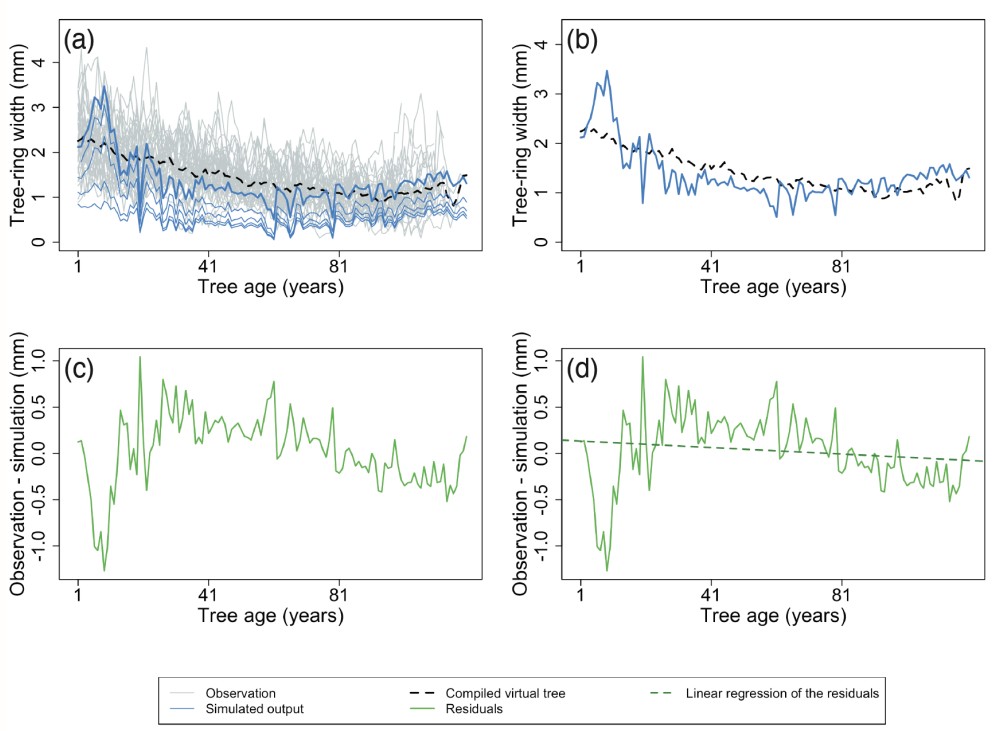

**Figure 5. Example of the major steps for calculating the metrics of the benchmark for the size-related trend in diameter increment.** The size-related trend in diameter increment can be assessed by calculating a time series for the average ring width after aligning the age of the individual trees (a, b). Observations are shown as grey lines and simulation as blue lines. The largest diameter class from the simulation is presented by the bold line. The black dotted lines represent the virtual tree based on the observations. The TRWs of this virtual tree are then subtracted from the simulated TRWs of the largest diameter class (c). Subsequently, a linear regression is used to quantify the temporal trend in the residuals (d). The green line denotes the model residuals and the green dotted line is the linear regression of the model residuals. Furthermore, the root mean square error (RMSE) between the simulations and observations is calculated (b; RMSE between blue line and black dotted line) and normalized by the length of time series to calculate the difference in observed and simulated growth trends. Observations and simulation are from a *Pinus sylvestris* site in Finland archived as finl052 (Meriläinen et al., 2004). For this example, calculated RMSE (b) is 0.39 (mm), and the slope of residuals (d) is -0.002 (mm/yr).

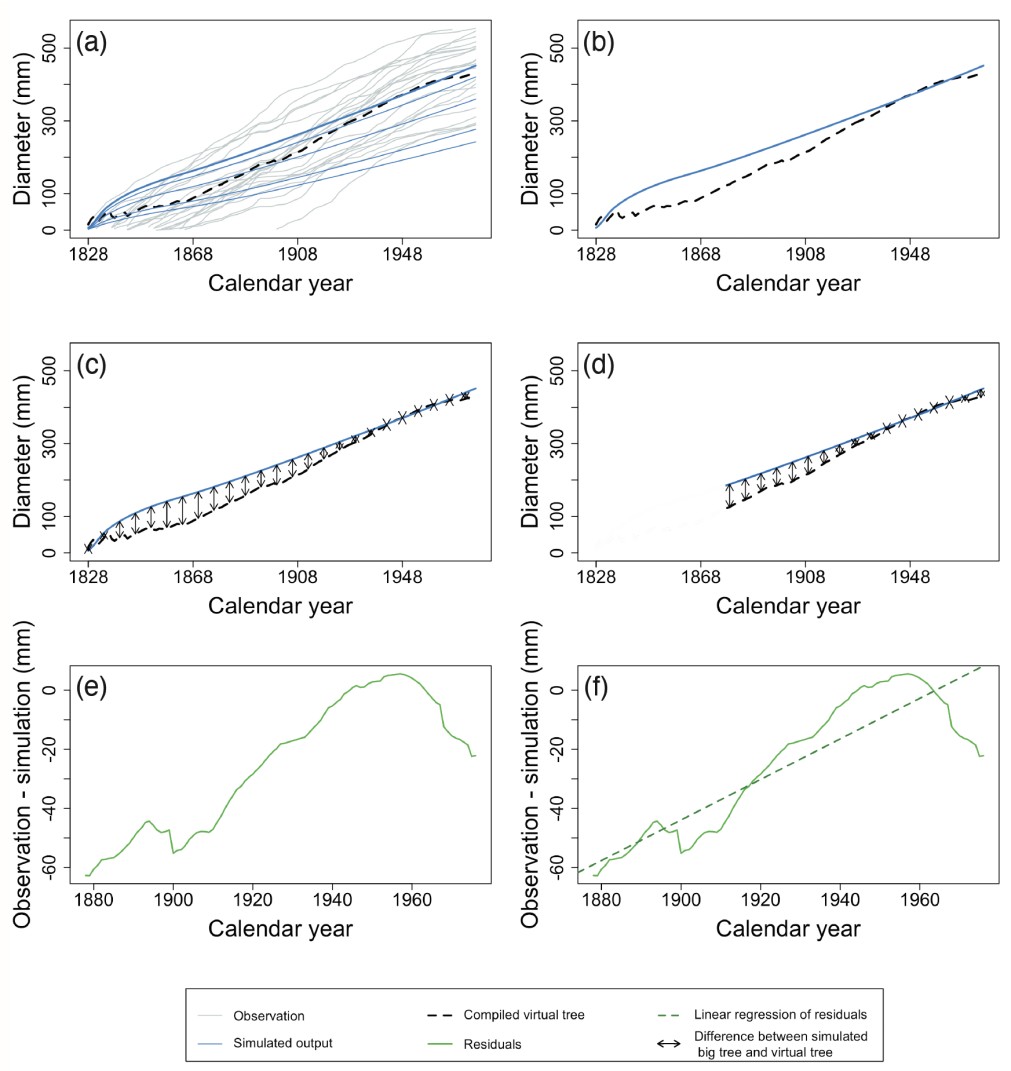

**Figure 6. Example of the major steps in calculating the metrics of the benchmark for the diameter increment in mature trees.** Individual tree records are ordered by calendar year and for each year the average observed diameter is calculated (a). Observations are shown as grey lines and simulation as blue lines. The largest diameter class from the simulation is presented by the bold blue line. Black dotted lines represent the yearly average of observations. Note that X-axis in Fig. 6 is different from Fig. 5. Under the assumption that the observed trees are the biggest trees from a given site, the virtual tree is compared with the largest diameter class from the model (b, c). Given that for the most recent decades both the fast and slow growing trees are still alive and could have been sampled, only the recent decades (ten decades, in this example) of the virtual tree growth are compared to the simulations (d). The RMSE (d; black arrows) and trend of the residuals between the virtual tree and the largest diameter class simulated are calculated (e, f). The x-axes of e, f zooms in on the selected period. The green line denotes the residuals and the green dotted line is the linear regression of the model residuals. Observations and simulation are from a *Pinus sylvestris* site in Scotland archived as brit021 (Schweingruber, 1995). In this case, RMSE (d) and the slope of residuals (f) were calculated as 33.65 (mm) and 0.68 (mm/yr), respectively.

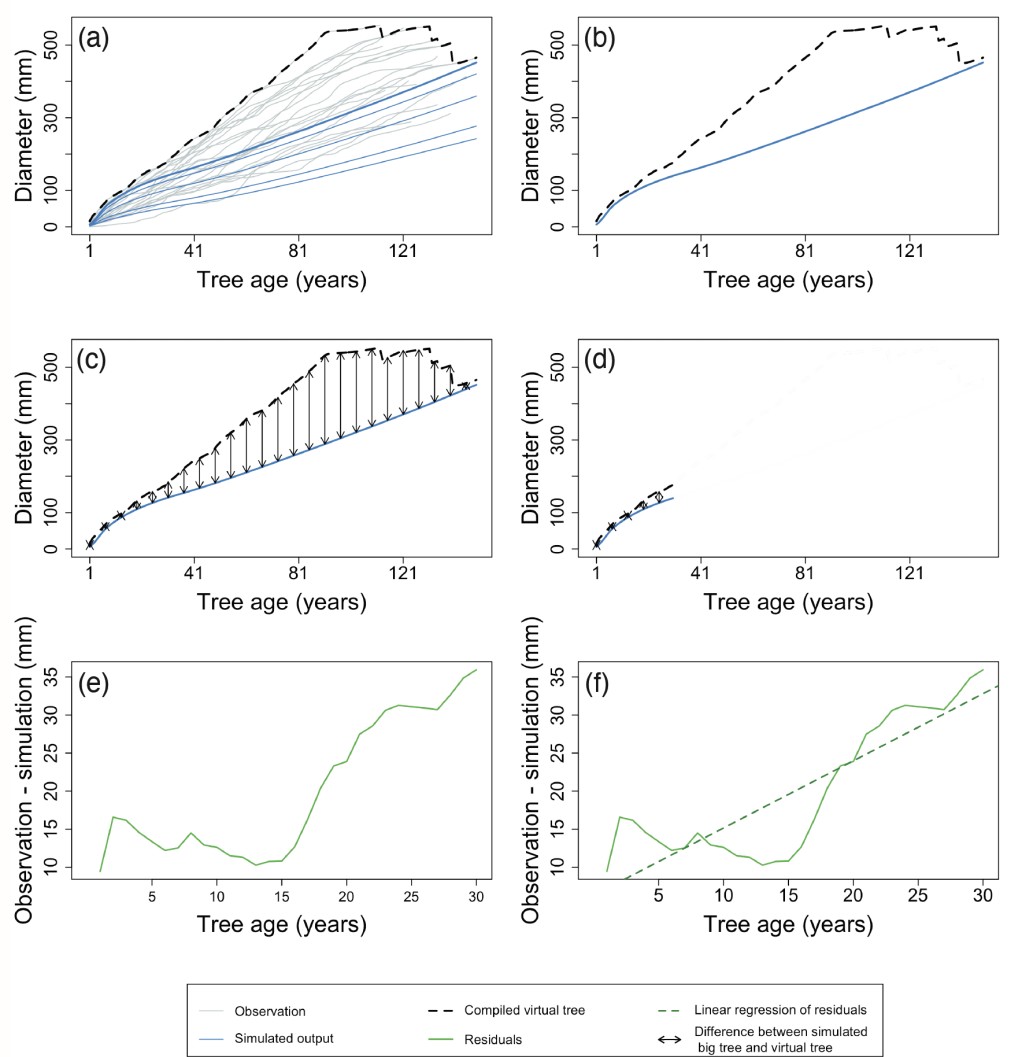

**Figure 7. Example of the major steps in calculating the metrics of the benchmark for diameter increment in young trees.** After aligning the TRW records of the individual trees by their age, a virtual tree is constructed by taking the maximum observed diameter of all trees for each year (a). Observations are shown as grey lines and simulation as blue lines. The largest diameter class from the simulation is presented by the bold line. Black dotted lines represent the yearly maximum of the observations. The growth of the virtual tree is then compared to the simulated growth of the largest diameter class (b) by calculating the RMSE (c) and trend of the residuals (e, f). The x-axes of e, f zooms in on the selected period, and the green line denotes the model residuals and the green dotted line is the linear regression of the model residuals. These calculations are limited to the first decades of the time series (d) to compensate for the bias caused by the fact that the old fast-growing trees died well before sampling took place. By using different approaches to evaluate the growth of young (this benchmark) and mature trees (the previous benchmark) the comparison accounts for the observation that the drivers of ring growth change when the trees grow taller (Cook, 1985). Observations and simulation are from *Pinus sylvestris* site in Scotland archived as brit021 (Schweingruber, 1995). The calculated RMSE (d) was 21.86 (mm) and the slope of residuals (f) was 0.88 (mm/yr) for this example.

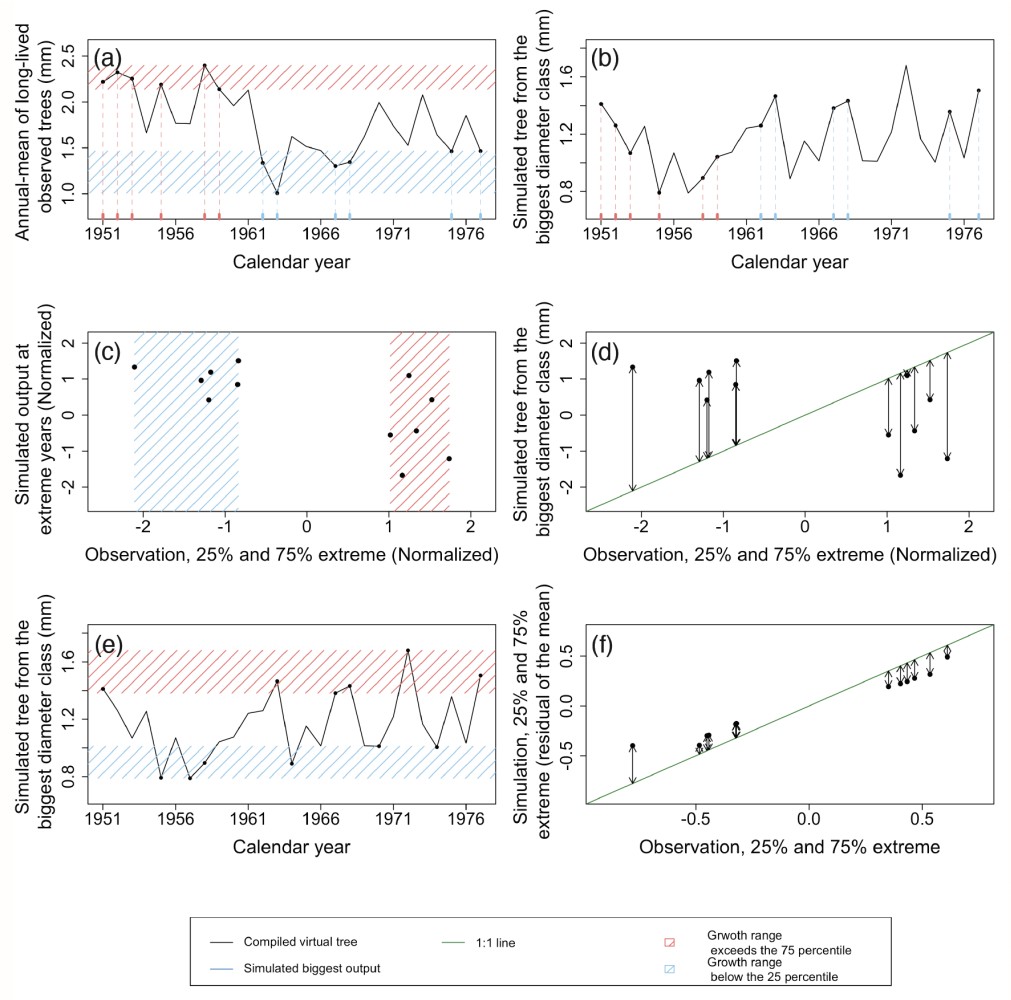

**Figure 8. Example of the major steps in calculating the metrics of the benchmark for extreme growth events.** In this benchmark, extreme growth is defined as the first and last quartiles in TRW ordered by calendar year and averaged over the individual trees records (a). Red shaded area and ticks represent observations exceeding the 75 percentile and blue shaded area and ticks represents observation below the 25 percentile (a). The TRW simulated for the largest diameter class are then extracted for the years identified in a (b). Both observations and simulations were normalized to remove the difference in the range of values between configurations. These normalized values correspond to the X and Y axis in (c) and (d) for observation and simulation, respectively. Subsequently, the similarity between simulations and observations was tested by calculating the distance from the 1:1 line (shown in green in d), which is equivalent to the RMSE for years with extreme growth (d). An additional metric is calculated in a similar way but by using both the 25% and 75% extreme values of the simulation and observation regardless of the year (e, f). This test identifies if the simulation can reproduce the amplitude of TRW. The observations and simulations were not normalized to assess the absolute amplitude. Possible uncertainties from using reconstructed climate forcing, were avoided by limiting the calculations of both metrics to the past five decades for which climate observations are available. Observations and simulation are from *Pinus sylvestris* site in Spain archived as spai006 (NOAA-WDS for Paleoclimatology, 2020). In this test case, RMSE for extreme years (d) was 0.57 (mm) and RMSE for extreme growth (amplitude; f) was 0.03 (scaled).

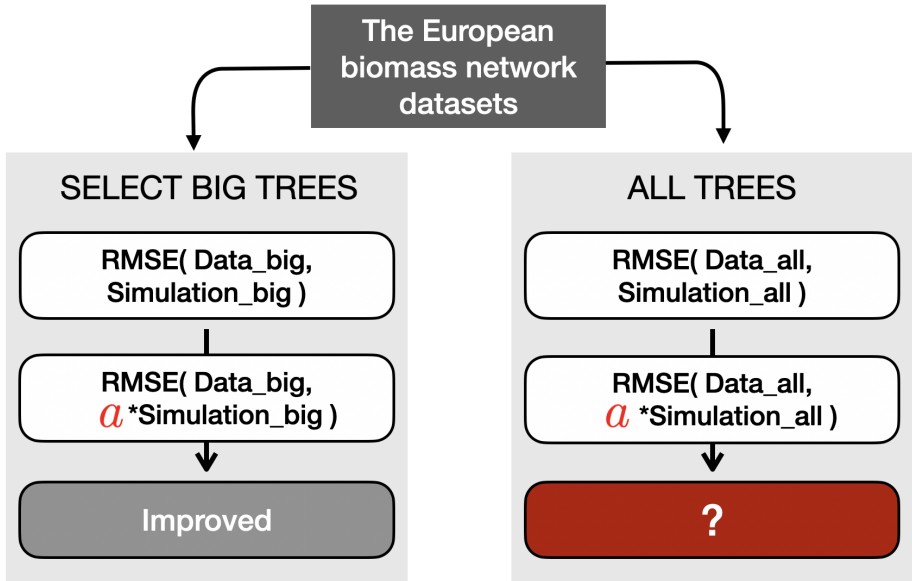

**Figure 9. Schematic representation of the verification process for the RMSE-metric.** Before the verification, two types of datasets were prepared: big-tree data (limited to the 15% biggest trees) and all-tree data. In this example, the simulated TRW was multiplied with a verified modifier such that it minimized the RMSE between the simulated and observed TRW for the 15% biggest trees (see Section 3.2 for details). The same multiplier was then applied to the all-tree data and the RMSE was calculated. Finally the decrease or increase in RMSE with the multiplier was compared to the RMSE obtained without the multiplier. The other two modifiers which are detailed in section 3.3 follow a similar approach.

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
