# Peer review of "Using the International Tree-Ring Data Bank (ITRDB) records as century-long benchmarks for global land-surface models"

_Geoscientific Model Development, 2020_

## Referee Comment (RC1) · Anonymous Referee #1 · 18 May 2020

In this manuscript, the authors aim to develop a method for benchmarking land surface models (LSMs) against tree ring data. The paper details the issues with using such data, specifically from the International Tree Ring Database (ITRDB) and describes a method for overcoming the biases in these data. This method is then tested using the ORCHIDEE model with four different levels of complexity. This is a valuable study as tree ring data is a resource largely untapped by the modelling community and which can give us insight into tree growth over long periods of time. I also think that the approach the authors have taken is very robust and should be applied in more model evaluation studies - the careful assessment of the biases in the data as well as the model features needed for a comparison with the data.

[Figure]

I believe that a lot of time and effort has gone into this study, however, I think it needs some editing to make it more clear and convincing.

**Major comments**

Having read the whole paper, it remains unclear to me if the method presented here actually works and in fact, how can we tell if it does work. I suspect that the answers to these questions lie in figure 4, but this figure is very hard to understand. First of all, it comes before the figures illustrating how each of the benchmarks works, so it is not clear to the reader what the different quantities in the figure are. It would help if figures 5-8 came first. Secondly, I am unsure about the comparison between the model and the BACI data here. As far as I understand, the model is run at different sites than the ones in the BACI data. Why is this and are the sites similar in terms of their climate and stand characteristics? Also, why does this figure only show coniferous sites? Is this an issue with the available BACI data? Would the results look similar for broadleaf deciduous sites?

The description of ORCHIDEE contains a lot of general details that I'm not sure are needed in a journal such as GMD e.g. capacity to be coupled/decoupled, variable grid size etc. On the other hand, I find the details of the model setup somewhat sparse, and these details are needed to understand the model results. A simple solution would be to just take the detailed description of the setup from the supplementary material and add it to the main text.

I'm not sure I understand why the four different models are needed, if the specifically stated purpose of this paper is not to evaluate the model. It adds an extra level of complexity that makes an already long and complicated paper even longer. If the application of the four model versions is insightful, it would help if this was discussed somewhere. From Fig. 9 it looks as if for some benchmarks the differences between sites are bigger than the differences between model versions - is this caused by climate, stand age, stand density?

[Figure]

The paper opens with a relatively long discussion about the issues and biases in using tree ring data to benchmark models. The problem is then that the actual discussion section largely repeats the same arguments. It would be more interesting to see here a discussion about the generality of the benchmarks - can they be used at different sites? With different species? For other models?

**Minor comments**

L 83 Is the assumption that forests are unmanaged likely to be correct?

L 84 How was the start year set to match observations? Is this based on inferred tree age from the tree ring data or is there more information on forest age?

L 80 Was there data on N deposition also used as forcing?

(Note line numbering appears to break after 100)

Figure 4 - I'm not sure what 'trend' refers to. The model value for 'young' does not appear to have error bars. Why does the data set have 27 sites and the model is run at 10 sites only? Are all these values for coniferous sites only?

L 130 These benchmarks are discussed earlier but only explained here

L 185 Are the results of the leave-one-out approach shown somewhere?

L209 (I think, p 17) I don't see why the dynamic leaf N in itself would cause a problem, as it is a realistic process. It is much more likely that having a more complex representation of N processes exposes an issue with other parts of the N cycle in the model (e.g soil)
* * *

---

## Referee Comment (RC2) · Anonymous Referee #2 · 29 May 2020

Jeong et al., proposed a new method to use ITRDB tree ring width to benchmark Land Surface Model (LSM) in the century-long period, to enable the benchmark could be extended back to those periods well before human-induced environmental changes. This creative way of using ITRDB for a longer-term benchmarking, transiting from pre-industrial to present-day environmental conditions over the past century, could be a very useful tool for model development and is very relevant for future predictions. Because it could potentially cover both the stable (pre-industrial) and fast (recent decades) climate change period. Four benchmarks, combined with the idea of the observed virtual tree are introduced to account for the sampling bias in ITRDB tree ring data and the fact that size-related growth exceeds the one caused by environmental changes. It

is very interesting to read and learn about how and why those metrics were chosen. The paper is well written and informative. I appreciate the huge and complicated work that the authors have done.

However, my biggest question or I hope to read from this paper is why and how this new approach works. For example, the data-based evidence is needed for why the size-related trend in diameter increment should be unique enough to be used as a character to distinguish different sites with different past century's climates. Why the diameter history, which contains not only the current year's growth signal but also carries previous years bias (possibly), was used to evaluate whether model performs well in diameter increment pattern in both young and mature trees? And the European regional case study didn't give a clear conclusion for the whole benchmarks.

There are a few minor queries, especially for the four benchmarks.

I am curious about whether the simulated ring width has been tuned before the final model run by adjusting some of the parameters. Could the authors be clear about whether there is the tuning process? And if so, the way of using RMSE or difference between observation and simulation can be tricky. Because those "artificial" bias could potentially have a big influence on such RMES-based benchmarks by simply changing/tuning the level of growth.

Figure 4: more details about what is compared are needed. Is y-axis the mean of ring width?

Figure 5: The exhibited slope estimation at Panel (d) looks not that convincing. The flat slope is heavily influenced by the big continuous underestimation for the young growth. And there is an obvious downward trend since the tree getting bigger. The slope estimation could make more sense (or be more robust) if data (difference) could be randomly arranged, not by age; or if it is not showing the consistent longer-term difference in either the positive or negative way for a certain period.

Figure 6: Details to explain how the "recent year" at Panel (d) was decided is needed? And would this "cut" of data scarify the length of data availability, considering this new methodology is targeting for "century-long" model-data comparison, and the mature tree is one of the more important benchmarks in the four?

Figure 7: Some logical reason why only the first few decades (30ish) years are chosen for this benchmarking is needed. The comparison was limited within the first few decades of the time series for young trees comparison. It was mentioned because the old fast-growing trees died well before sampling took place. But actually, those "young" fast-growing trees lived through a much longer period shown in Panel (a).

Figure 8: It looks like the extreme event benchmark is the most climate-sensitivity related benchmark. However, the period is limited for the most recent years when the most reliable observed climate data is available, which is not consistent with the other three benchmarks. This somehow downsized the importance of this new benchmarking method. (Because the longer-term benchmark is one of the major breakthroughs.) Does this mean the other three benchmarks are not that sensitive to the quality of the climate data, especially to the climate variations? The extreme value was extracted from the average of the observation, without any size related detrending. Would the size-related growth have any impact on the quantile statistic? Panel (d): "mm" in y-axis title should be "Normalized". Panel (f): how different years' value were matched if only the quantile was applied for both observation and simulation? Is there any explanation about why the model is always overestimating the growth for both the good and bad years. Is it because the original value of TRW (not the standardized one) was used. Again, I am wondering whether there is a modelling tuning process to adjust the simulated ring width closer to the observation. I understand Panel (e) and (f) is to test the ability to reproduce the amplitude of TRW, which has also been majorly targeted by the former three benchmarks. However, it might also logically make sense by simply using the normalized value if the above three benchmarks passed. Meanwhile, relative change can be more relative to climate sensitivity comparison, if the simulated growth
was tuned.

Table 1: Wider space between each row of the table could enhance the readability.

[Figure]

---

## Author Comment (AC1) · 8 Jul 2020

We would like to thank the reviewers for their time and effort in reviewing our manuscript. The review comments pointed us to parts of the manuscript that could be simplified and parts that need additional clarification. We are confident that we can address the vast majority of the review comments in a revised manuscript. Addressing the referee comments will require simulations at new and more sites, preparing new figures, and analyzing new results and will thus result in major revisions (see below).

**Major discussion**

Referee #1

Having read the whole paper, it remains unclear to me if the method presented here actually works and in fact, how can we tell if it does work. I suspect that the answers to these questions lie in figure 4, but this figure is very hard to understand. First of all, it comes before the figures illustrating how each of the benchmarks works, so it is not clear to the reader what the different quantities in the figure are. It would help if figures 5-8 came first. Secondly, I am unsure about the comparison between the model and the BACI data here. As far as I understand, the model is run at different sites than the ones in the BACI data. Why is this and are the sites similar in terms of their climate and stand characteristics? Also, why does this figure only show coniferous sites? Is this an issue with the available BACI data? Would the results look similar for broadleaf deciduous sites?

We agree with the major comments of referee #1 concerning the complexity of the manuscript. Our initial idea was to use the BACI data to show that the assumptions required to use the ITRD data in model evaluation are acceptable. We then applied the model on the ITRDB data as a case study. The referee comment made us realize that the ITRDB simulations are not necessary to demonstrate the validity of the proposed benchmarks and we therefore propose to limit the analysis strictly to the BACI data. In a revised manuscript the validity of the proposed benchmarks could be tested by using 1) the BACI data and ORCHIDEE simulations for all trees; and 2) the BACI data and ORCHIDEE simulations for the biggest trees. Such an analysis could demonstrate that ITRDB data which typically contain the biggest trees in a stand contain useful information about the entire stand and can therefore be used for model development and evaluation. It is clear that this change in model experiment will result in many changes in the manuscript, likely including a more profound verification and discussion of the proposed benchmarks. Because the study will only use the BACI data, we hope we will be able to present simulations for both deciduous and conifer stands without making the results and discussion overly complex. We think this new and simplified approach will better illustrate the strengths and weaknesses of the proposed benchmarks and will better demonstrate which of the proposed benchmarks can be used with ITRDB data and which should not.

I'm not sure I understand why the four different models are needed, if the specifically stated purpose of this paper is not to evaluate the model. It adds an extra level of complexity that makes an already long and complicated paper even longer. If the application of the four model versions is insightful, it would help if this was discussed somewhere.

The four model configurations are a leftover of the initial test but the referee is right in questioning their need for the purpose of this study. A revised manuscript will only report the configuration labelled "Ndyn" which is now the ORCHIDEE default.

Referee #2

My biggest question or I hope to read from this paper is why and how this new approach works. For example, the data-based evidence is needed for why the size-related trend in diameter increment should be unique enough to be used as a character to distinguish different sites with different past century's climates. Why the diameter history, which contains not only the current year's growth signal but also carries previous years bias (possibly), was used to evaluate whether model performs well in diameter increment pattern in both young and mature trees? And the European regional case study didn't give a clear conclusion for the whole benchmarks.

We read this comment as an inquiry about the foundation of dendrochronology and its value for benchmarking models. The foundation of dendrochronology rests on the observation that at the site-level trends in tree-growth contain valuable information about the ontogenetic growth during establish and endogenous competition from canopy closure. If the tree-ring record is long enough a single stand may have experienced different environmental conditions. From a modelling point of view the first challenge is to simulate the response in tree growth to these environmental changes. A second challenge that could help to achieve the first is to simulate with a single model and a single set of parameters, the growth from sites which experienced different environmental changes. From this point of view, changes in diameter growth due to environmental changes can be used to benchmark models (but it requires that size related growth trends are accounted for). Simulating growth trends should therefore be prioritized over matching the endpoints in diameter.

We think the questions of referee #2 are the result of a too concise introduction in the manuscript about the prerequisite of tree-ring research. The revised manuscript will elaborate more on this context. Also, we think that the new approach that will be used to address the concern of referee #1 will help to clarify the first question of referee #2: why and how the proposed benchmarks (don't) work.

Figure 5: The exhibited slope estimation at Panel (d) looks not that convincing. The flat slope is heavily influenced by the big continuous underestimation for the young growth. And there is an obvious downward trend since the tree getting bigger. The slope estimation could make more sense (or be more robust) if data (difference) could be randomly arranged, not by age; or if it is not showing the consistent longer-term difference in either the positive or negative way for a certain period.

The suggestion from referee #2, which is randomizing the residuals of the tree-ring trend, is creative but we don't understand how it could overcome over interpretation. The current analysis shows no trend for the ordered residuals with as the referee noted years of underestimation followed by years of overestimation. If we would randomized such residuals we would most likely find no trend in the residuals. If we misunderstood the proposal of the referee, we are open to adjust the statistics following new instructions. It should be noted that for this study, the trend in tree-ring width contains the information we are seeking to use as it

quantifies the change of tree growth with time. The information gets its importance from anthropogenic-driven long-term environmental changes, for instance, an increase in CO2 and nitrogen deposition. Randomization would break the growth trend over time and would hide the information we are looking for. The lack of a slope in Fig. 5 (d) was mostly driven by the trend for the established stages and implies that the model mimicked the trend relatively well. Note the benchmark combines the slope (Y-axis) with an RMSE (x-axis). The absence of a slope with a high RMSE suggests that there are is substantial over and underestimation of the trend. A good model is expected to result in a zero slope and a low RMSE. In the revised manuscript we try to better explain why the benchmark targets the trend in growth.

Figure 8: It looks like the extreme event benchmark is the most climate-sensitivity related benchmark. However, the period is limited for the most recent years when the most reliable observed climate data is available, which is not consistent with the other three benchmarks. This somehow downsized the importance of this new benchmarking method. (Because the longer-term benchmark is one of the major breakthroughs.) Does this mean the other three benchmarks are not that sensitive to the quality of the climate data, especially to the climate variations?

We largely agree with the referee but this comment made us realize that we need to improve the flow and presentation of the manuscript. We are indeed looking for a long-term benchmark because those are rare. Tree ring records go back far enough in time so we selected these records. We then looked which known issues with the ITRDB data should be taken into account such that these data could still be used (some of the known ITRDB issues can only be addressed by adjusting the model. This process resulted in four possible benchmarks). Two out of the four proposed benchmarks are long-term. We agree with the referee that the time horizon of the extreme event benchmark is not long enough to qualify as long term. We would not conclude this downsizes the importance of the new method. Rather our study proposes four different benchmarks that could be used to evaluate different aspects of simulating historical growth. In a revised manuscript we will try to better clarify this reasoning and we will improve the current table 1 such that it systematically describes the characteristics of each benchmark and discusses the implications for model evaluation.

**Specific comments**
Specific comments, for example, line numbers, the structure of the discussion, the order of the figures, figure caption and tables will all be addressed in the revised manuscript. The remaining specific comments are discussed below

Referee #1
From Fig. 9 it looks as if for some benchmarks the differences between sites are bigger than the differences between model versions - is this caused by climate, stand age, stand density?
The large variation across sites is indeed expected to come from climate, stand density but also nitrogen and water availability. This variation is one of the main reasons why for large-scale models multi-site parameter optimization is preferred rather above tuning the parameters against a single site. ITRDB could provide multi-site observations for the same PFT but from

locations with rather different climates. Given that only a single model version will be shown in the revised manuscript this comment will not be addressed in the revised manuscript.

We are trying to select site that are 150 years or older. Given the age and the location of the sampled forest, we think that such sites are unmanaged or experienced little management. Even in heavy managed regions as Europe such unmanaged forest fragments often less than a 1ha are abundant especially in the mountain regions. Although the revised manuscript will focus on BACI data for which the management status of the sites is better documented than for the ITRDB we will add this assumption in revised manuscript as the purpose of the manuscript is to propose benchmarks that could be used with the ITRDB data.

ITRDB doesn't have metadata for forest age so start and end year was matched to the length of the longest observation. The same applies to the older stands in the BACI data set. We will add this assumption in the revised manuscript.

ORCHIDEE r5698 does use a global N deposition map based on the ongoing CMIP6 efforts. We will add a short description of these data in the revised manuscript.

The error-bar in Fig. 9 is the outcome of the leave-one-out approach. This was not reported in the manuscript. We will add it in the revised manuscript.

We agree that dynamic leaf N itself is not a problem. Rather, because leaf N is dynamic in ORCHIDEE it could be overestimated especially if the optimal value, which is prescribed for each PFT. Future model use, evaluation and optimization may help us to establish whether this is indeed a very sensitive model parameter. We will rephrase this paragraph of the manuscript to avoid similar confusion from reading the revised manuscript.

**Referee #2**

During model development data from ten ITRDB sites (aust112, cana106, chin037, finl055, fran4, id007, japa011, mo009, nepa003, spai055, and turk027) were used to assess the impact of the developments and to search for the most sensitive parameters. No formal model tuning took place because that is the objective of a follow-up study. We will clarify this issue in the revised manuscript by adding this information. The purpose of this manuscript is to show which benchmarks could be used such that ITRDB data could be used as one of the data streams that

is routinely used in model parameterization. This paper does not focus on how well/poorly ORCHIDEE can simulate tree rings. We see this as a two-stage process: (1) can the data be used and how, and (2) can ORCHIDEE be used?

Figure 4: more details about what is compared are needed. Is y-axis the mean of ring width?
Fig 4 compares the benchmarks. In the revised manuscripts all figures showing results will change given the initial comment of referee 1. While preparing the revisions we will pay special attention to improve the caption of the substitute of Fig 4.

Figure 6: Details to explain how the "recent year" at Panel (d) was decided is needed? And would this "cut" of data scarify the length of data availability, considering this new methodology is targeting for "century-long" model-data comparison, and the mature tree is one of the more important benchmarks in the four?
The 50-year cut off is somehow arbitrary but most of series considered during this study show a slowly decreasing growth-trend after 50 years. Also, the 30-year cut off used in Figure 7, is arbitrary. Here 30 years instead of 50 years were chosen because for most time series considered the first 30 years are characterized by fast changes in tree growth. The benchmarks target time series of 150 years and longer because those trees experienced considerable environmental changes. If the first 50 years are cut, the time series still contains 100 years during which $CO_2$, temperature, precipitation and nitrogen availability may have changed. We will add this reasoning in the revised manuscript.

Figure 7: It was mentioned because the old fast-growing trees died well before sampling took place. But actually, those "young" fast-growing trees lived through a much longer period shown in Panel (a).
For consistency reasons, the same site was used to illustrate each of the four benchmarks (Fig. 5-8). The referee is correct that this approach may confuse readers. The site that was selected represents a relatively even-aged forest. In the revised manuscript we will chose a good example to illustrate each benchmark. This implies that consistency will be traded for readability.

Would the size-related growth have any impact on the quantile statistic?
We do indeed expect that the most dynamic phases of size-related growth could have an impact on the quantile statistics. For that reason, we select sites that already passed the dynamic phase in the year we start the analysis. This means that for this specific benchmark, trees should be 50 years or older in 1950 (the year after which we expect the climate reconstructions to become more reliable). By doing so we expect that there is almost no age-related growth trend in the years that are considered in the benchmark (1950-present). We will try to better clarify this line of reasoning in the summary table proposed to address the previous comment (improved version of the current table 1).

Panel (f): Is there any explanation about why the model is always overestimating the growth for both the good and bad years. Is it because the original value of TRW (not the standardized one) was used.
We agree with the hypothesis of the referee. The target is the absolute value of tree-ring width, and this shows the model overestimates overall tree-ring widths for this site. We plan to add

directions on how each benchmark can be used for evaluating models. Because Fig. 8 is drawn to explain the extreme benchmark, we omitted explanations about model performance.

I understand Panel (e) and (f) is to test the ability to reproduce the amplitude of TRW, which has also been majorly targeted by the former three benchmarks. However, it might also logically make sense by simply using the normalized value if the above three benchmarks passed. Meanwhile, relative change can be more relative to climate sensitivity comparison, if the simulated growth was tuned. In this study we use amplitude as the difference between the lowest and highest observed diameter increment. According to this definition the fourth proposed benchmark is the only benchmark that is considering the amplitude, all other benchmarks are considering the trend in diameter growth. In the early years this trend is caused by ontogeny in later years the climate sensitivity could become more pronounced in the tree ring records. We will try to better clarify each benchmark in an improved version of the current table 1.

---

## Author Response (AR1)

We'd like to thank to referees for their comments as they helped us to greatly simplify the manuscript and improve its key messages.

This document contains a point-by-point response to the referees.

**Referee #1**

**Major comments**

Having read the whole paper, it remains unclear to me if the method presented here actually works and in fact, how can we tell if it does work. I suspect that the answers to these questions lie in figure 4, but this figure is very hard to understand. First of all, it comes before the figures illustrating how each of the benchmarks works, so it is not clear to the reader what the different quantities in the figure are. It would help if figures 5-8 came first.

The description of benchmarks was moved to the background section (Section 2.3) so it comes earlier in the manuscript. Fig 4 was indeed overly complex and has been replaced by a different analysis presented in table 2. This change in focus also helped to better demonstrate which of the benchmarks can be used with ITRDB data and which benchmarks require data from an unbiased sampling design.

Secondly, I am unsure about the comparison between the model and the BACI data here. As far as I understand, the model is run at different sites than the ones in the BACI data. Why is this and are the sites similar in terms of their climate and stand characteristics? Also, why does this figure only show coniferous sites? Is this an issue with the available BACI data? Would the results look similar for broadleaf deciduous sites?

We understand where this confusion comes from and largely simplified the study by only using the BACI data. Rather than using ITRDB, the BACI data were resampled to mimicked ITRDB-type of sampling bias. The data from the unbiased and biased sampling designs were then used to benchmark. Differences in conclusions based on the data from the biased and unbiased sampling designs suggest that the benchmark is sensitive to the sampling design. This comment was instrumental in the revisions as it helped to simplify the manuscript as a whole and at the same time better demonstrates which of the proposed benchmarks can be used with ITRDB data. Within all BACI sites, sites dominated by a single coniferous species (spruce) and a single deciduous species (beech) were analyzed (Section 3.2) in response to the referees' question.

The description of ORCHIDEE contains a lot of general details that I'm not sure are needed in a journal such as GMD e.g. capacity to be coupled/decoupled, variable grid size etc. On the other hand, I find the details of the model setup somewhat sparse, and these details are needed to understand the model results. A simple solution would be to just take the detailed description of the setup from the supplementary material and add it to the main text.

Details about the model setup were necessary for the previous manuscript to understand the 4 different configurations better. But, considering the referee #1's comment, the ITRDB test

cases were removed, and the new test uses only one configuration (called 'Ndyn' previously). The rest of the setup is described in Section 3.3.

I'm not sure I understand why the four different models are needed, if the specifically stated purpose of this paper is not to evaluate the model. It adds an extra level of complexity that makes an already long and complicated paper even longer. If the application of the four model versions is insightful, it would help if this was discussed somewhere. From Fig. 9 it looks as if for some benchmarks the differences between sites are bigger than the differences between model versions - is this caused by climate, stand age, stand density?

These tests were a left-over from the initial work on simulating tree ring widths with ORCHIDEE. The referee's comment made us realize that they were no longer needed and that we could bring the same message with a much simpler simulation set-up. We removed the different model configurations from the manuscript which enables us to better focus on examining the proposed benchmarks (Section 3.3 and 3.4).

The paper opens with a relatively long discussion about the issues and biases in using tree ring data to benchmark models. The problem is then that the actual discussion section largely repeats the same arguments. It would be more interesting to see here a discussion about the generality of the benchmarks - can they be used at different sites? With different species? For other models?

The whole result section and half of the discussion were rewritten in line with the major revisions we made in the study design. This rewrite includes more details on the validity and capability of the proposed benchmarks when using ITRDB data.

**Minor comments**

L 83 Is the assumption that forests are unmanaged likely to be correct?
We got the management status of each BACI site. This was applied to each simulation (Table S2).

L 84 How was the start year set to match observations? Is this based on inferred tree age from the tree ring data or is there more information on forest age?
The start and end year were matched to the length of the longest observation. This has been clarified in the revised manuscript (Added to L377).

L 80 Was there data on N deposition also used as forcing?
This information is added to L343-344.
(Note line numbering appears to break after 100)
We adjusted the page margin.

Figure 4 - I'm not sure what 'trend' refers to.
We added the paragraph describing benchmarks further in the discussion (L491-523) and modified Table 1.

The model value for 'young' does not appear to have error bars.
Because 'young' was not affected much by the leave-one-out approach, but this figure is not used anymore.

Why does the data set have 27 sites and the model is run at 10 sites only? Are all these values for coniferous sites only?
This was because the previous figure 4 compared BACI datasets with ITRDB simulation. This issue has been resolved by simulating only at BACI sites. This resulted in full consistency between the numbers of observations and simulations.

L 130 These benchmarks are discussed earlier but only explained here
This part has been moved to the background section (Section 2.3).

L 185 Are the results of the leave-one-out approach shown somewhere?
It was missed in the previous version but, since the result presentation has been changed, this comment was not applied.

L209 (I think, p 17) I don't see why the dynamic leaf N in itself would cause a problem, as it is a realistic process. It is much more likely that having a more complex representation of N processes exposes an issue with other parts of the N cycle in the model (e.g soil)
Since the main result and presentation was changed to focus on the verification of the benchmarks, the content related to evaluating the model was removed.

**Referee #2**

my biggest question or I hope to read from this paper is why and how this new approach works. For example, the data-based evidence is needed for why the size-related trend in diameter increment should be unique enough to be used as a character to distinguish different sites with different past century's climates. Why the diameter history, which contains not only the current year's growth signal but also carries previous years bias (possibly), was used to evaluate whether model performs well in diameter increment pattern in both young and mature trees?
This comment in combination with several questions from referee #2, made us realize that the explanation for the proposed benchmarks and what they are targeting was too concise in the previous manuscript. A large effort was made to clarify these issues by simplifying the study design as well as by adding further explanation about benchmarks, especially in relation to the application for the model evaluation (L491-523).

And the European regional case study didn't give a clear conclusion for the whole benchmarks.
Now the manuscript doesn't use ITRDB for simulations, but focuses on verifying the proposed benchmarks by using bias-free datasets. Unfortunately, the datasets are collected mostly in Europe, so the simulations are still limited to Europe. However, we believe it is a better way to evaluate the proposed benchmarks than simulating ITRDB sites over different continents.

There are a few minor queries, especially for the four benchmarks.

I am curious about whether the simulated ring width has been tuned before the final model run by adjusting some of the parameters. Could the authors be clear about whether there is the tuning process? And if so, the way of using RMSE or difference between observation and simulation can be tricky. Because those "artificial" bias could potentially have a big influence on such RMES-based benchmarks by simply changing/tuning the level of growth.

The model was manually adjusted against ITRDB sites (thus not against BACI sites) during model development. This information has been added in the revised manuscript Section 3.3.

Figure 4: more details about what is compared are needed. Is y-axis the mean of ring width?

The y-axis was the ratio of the difference in benchmarks, however, Figure 4 has been removed and replaced by a different verification approach.

Figure 5: The exhibited slope estimation at Panel (d) looks not that convincing. The flat slope is heavily influenced by the big continuous underestimation for the young growth. And there is an obvious downward trend since the tree getting bigger. The slope estimation could make more sense (or be more robust) if data (difference) could be randomly arranged, not by age; or if it is not showing the consistent longer-term

We did not fully understand this comment. It was unclear how randomizing the trend can improve data processing. The trend in tree-ring width or diameter contains the information the model is expected to reproduce and the flat slope in Fig. 5 (d) implies the model simulated the trend relatively well. We interpreted this comment as a sign that we did not well explain this in the initial manuscript and used the revision to provide more explanation on these issues (L493-499, L519-523, and Table 1).

Figure 6: Details to explain how the "recent year" at Panel (d) was decided is needed? And would this "cut" of data scarify the length of data availability, considering this new methodology is targeting for "century-long" model-data comparison, and the mature tree is one of the more important benchmarks in the four?

We added the underlying reasoning in the revised manuscript (L264-267)

Figure 7: Some logical reason why only the first few decades (30ish) years are chosen for this benchmarking is needed. The comparison was limited within the first few decades of the time series for young trees comparison. It was mentioned because the old fast-growing trees died well before sampling took place. But actually, those "young" fast-growing trees lived through a much longer period shown in Panel (a).

We changed the example site in Figure 5 and 6, finl052 to brit021.

Figure 8: It looks like the extreme event benchmark is the most climate-sensitivity related benchmark. However, the period is limited for the most recent years when the most reliable observed climate data is available, which is not consistent with the other three benchmarks. This somehow downsized the importance of this new benchmarking method. (Because the longer-term benchmark is one of the major breakthroughs.)

Does this mean the other three benchmarks are not that sensitive to the quality of the climate data, especially to the climate variations?

Extreme even benchmark was built to evaluate plant growth sensitivity to year-to-year climate which requires more reliable climate data than the other benchmarks. Two of the other benchmarks quantifies long-term growth. We agree with the referee #2's opinion about the time frame of extreme benchmark, however, we think the complementary role of four different benchmarks can relieve the issue. As described above, the revised discussion and table 1 were improved to better clarify these issues.

The extreme value was extracted from the average of the observation, without any size related detrending. Would the size-related growth have any impact on the quantile statistic?

Only trees that passed the most dynamic phases of the size-related growth trend were considered in this analysis. The referee is right in asking this question as we forgot to provide this information in the previous manuscript. The revised manuscript explicitly mentions this selection criterion on L279.

Panel (d): "mm" in y-axis title should be "Normalized".

This comment has been applied to Table 2.

Panel (f): how different years' value were matched if only the quantile was applied for both observation and simulation? Is there any explanation about why the model is always overestimating the growth for both the good and bad years. Is it because the original value of TRW (not the standardized one) was used.

This was because the model overestimate overall tree-ring widths, however, the detailed description about the model evaluation since the test was removed.

Again, I am wondering whether there is a modelling tuning process to adjust the simulated ring width closer to the observation. I understand Panel (e) and (f) is to test the ability to reproduce the amplitude of TRW, which has also been majorly targeted by the former three benchmarks. However, it might also logically make sense by simply using the normalized value if the above three benchmarks passed. Meanwhile, relative change can be more relative to climate sensitivity comparison, if the simulated growth was tuned.

In this study we use amplitude as the difference between the lowest and highest observed diameter increment. According to this definition the fourth proposed benchmark is the only benchmark that is considering the amplitude, all other benchmarks are considering the trend in diameter growth. We added more information in Table 1 to distinguish the long-term and short-term benchmark.

Table 1: Wider space between each row of the table could enhance the readability.

The cell margin was increased for better readability.

[revised manuscript text omitted]

---

## Referee Report (RR1)

**Review Report**

This manuscript suggests comparison method to use the International Tree-ring Data Bank (ITRDB) for validating land surface models (LSMs). Authors actually adapted these methods to the output of the ORCHIDEE-LSM. As authors mentioned, data in the ITRDB has systematic biases, direct comparison between ITRDB data and outputs of LSMs should be inappropriate. So, author's motivation for developing some comparison method makes sense.

However, the way authors explain the methods is not enough. Also, in my sense, it is too much complicated. Honestly, I had quite hard time to read this manuscript, but I still cannot understand it all. So, the purpose of this study is right, but how to prepare the manuscript has significant problem.

**Major specific comments:**

(1) Line 206 "virtual forest"
No definitions for the "virtual forest". As authors mentioned in the line 229-230, the proposed method largely relies on the concept of "virtual forest". So, missing its definition is ridiculous.

(2) Line 216-220 "Calendar year aligned virtual forest"
I strongly doubted this can be a metric to be compared with outputs of LSMs, because it should have systematic and strong bias as follows.
At the early stage of year, it has low values, because sampled trees (old and big trees) are in its young stages. During the middle period of the data, new trees (at young stage) add to the average calculation, and these trees function as a burden to increase the metric value. During the late period of the data, no tree enters to the average calculation anymore, so the metric value should show intense increasing trend.

(3) Lines 141-142
This sentence concludes the paragraph. But, I cannot understand why this conclusion comes out.

(4) Lines 144-153
Is it possible to rewrite this paragraph so that readers, who are not familiar with

dendrochronology and LSMs, can easily understand? Honestly, I still do not understand the point of this paragraph.

(5) Line 241

Here, the "Size related diameter growth" is identical with the "Age-aligned TRWs" in line 212? Inexplicit rephrasing causes confusion, and hence should be avoided.

(6) Lines 253-274

I cannot follow logics here.

**Minor specific comments:**

(1) Lines 200-202

Citing figure 4 in these sentences does not make sense.

(2) Figure 1

I think this figure is needless.

(3) Figure 4a

What the vertical axis means? Tree-ring width of the outmost stratum?

(4) Figure 4c

According to the figure caption, lines on this figure should be identical to those of the figure 4b; Figure 4b and 4c only differ in the alignment on the x-axis. But, shape of the black-dotted-lines differ between these figures.

(5) Line 676

A typo exists. Location of the period would be immediately after "(c)".

(6) lines 677-678

Is this sentence an explanation for the figure 5b? But, figure 5b shows extraction of major lines on the figure 5a, not the root-mean-square error.

---

## Editor Decision (ED1)

**Specific comments**

1) L80: Reference to Figure 1: In Fig. 1 you first mention the virtual trees, which have so far not been defined. The caption needs to ensure that the figure can be understood. Add a brief statement on the virtual tree concept and where more information can be found in the manuscript. In addition, the term "maximum virtual tree" is not explained anywhere and only used in Fig. 3 besides here.

2) L145-L170: Same issue as above: Fig. 3 is referenced, which requires understanding the concept of virtual trees but virtual trees have not been introduced to the reader and there is no information where their definition can be obtained.

3) Caption of Fig. 3: "maximum virtual tree" not clear.

4) Section 2.2: I recommend to insert a separate (sub-)section on the virtual trees or at least mention the term in the section title, which would make it easier to point readers to this. I.e., rename section 2.2 to e.g. "...and addressing them using a concept of virtual trees" and/or split this section into (a) challenges and (b) the solutions proposed in this manuscript.

5) L201/L216 etc.: remove "across many sites". An ideal model would have this behavior across all sites.

6) Section 2.3 (ii) and Fig. 5: How is the cut-off time period (last ten decades) determined? Section 2.3 (iii) has a thirty year time period and it is mentioned that this may be arbitrary.

7) L373: I don't understand the reference to Fig. 9 here. Wouldn't Tab. 1 be more appropriate to show the 88 test cases?

8) L380: 72% is not correct anymore as apparently results for site SOB have changed from 0.001 to -0.001 since the last version of the manuscript, which would increase the rate to 77%.

9) L386: Aren't there additional sites that have a positive change?

10) Figure 7 legend: "Simulated biggest output" not clear

11) Supplement L348: Table S3 does not provide information on "issues of the ITRDB"

12) Supplement L384: "longer" -> "older"?

13) Supplement L388: "XX mm"?

14) Supplement L412: "biggest simulation" not clear -> "largest simulated diameter class"?

---

## Author Response (AR2)

Dear editor,

We would like to thank you and the referee(s) for the constructive comments and much appreciate your help and patience. Despite the considerable experience of the author team, we seem to have been struggling to clearly articulate the thinking behind this study. We have revised the manuscript according to your and the referee's comments and suggestions (below is a point-by-point response). It is our hope that this study may help to end an era in which the use of the ITRDB data for model benchmarking was often discarded based on rather superficial reasoning. In this study, we aim to show that the ITRDB data contain information that can be used to benchmark land surface models.

Kind regards,

Jina Jeong on behalf of the author team

Referee
* * *
**Why the optimisation is needed**

The verification approach is explained in its own section and in that section the optimization is explained (L394-399 and L408-417). This explanation is supported by Fig 9. In a previous version of the text we used the term "optimization" in the context of the verification. We realize that this may have added to the confusion because "parameter optimization" (aka "tuning") in widely used in the modelling community. The approach we applied in the verification does not touch (or tune) any of the model parameters but instead modifies the outcome (in a transparent way), hence, we now use the term "modifiers". The text now reads: "The verification required three additional steps (Fig. 9): 1) The simulated TRW from the largest diameter class were transformed by modifiers (see below) to minimize the two metrics of each benchmark ; 2) the same modifiers were then applied to all simulated diameter classes both metrics for all four benchmarks were calculated using the all-tree data, and 3) the actual verification tested whether for a given metric and a given benchmark the modifier improved for the big-tree sample as well as the all-tree data. Improvement of a specific metric of a benchmark was quantified by subtracting the pre-modified value for that metric from its post-modified value for the all-tree data. A negative value thus indicated an improvement. If this was the case, the benchmarks of the big-tree and all-tree data were said to be consistent, implying that using this benchmark in combination with the ITRDB data would reveal the same model shortcomings as benchmarking ORCHIDEE against TRW data from all-tree networks. Across the 11 sites and for each of the four proposed benchmarks, sites where the test improved for both datasets were counted to estimate the confidence in using ITRDB in benchmarking LSMs."

*If the agreement between the two datasets was tested for statistically in any way.*

This question comes with a nuanced answer. Each individual benchmark is based on well-established statistics (RMSE and regressions) and the modifier is embedded in well-established statistics (RMSE). The change in RMSE and regression as shown in Table 2 is not subject to a statistical test. Note that the objective of comparing the datasets is not to establish whether there are significant differences but to establish whether the direction of the conclusions (e.g., an over or underestimation) based on ITRDB would similar to the direction of the conclusions based on the biomass network. If the direction of the conclusions is the same, their significance has little to no meaning in this context.

*I am wondering if it is possible to show plots of the bias and the slope (like in figures 4-7, last panel) for a few sites,*

Fig. 10 was added to the main texts and shows exactly the details of the benchmarks for four (out of 11) individual sites. We show 2 deciduous and 2 conifer sites and selected these sites randomly (first sites after sorting them in alphabetical order).

Editor
* * *
*it remains unclear in how far the new benchmark methodology is applicable. I agree that more information and a clear presentation is required to show this. For example, the optimizations and model parameter adjustments are challenging to interpret in the context of applying and benchmarking LSMs. E.g. L375 states that ORCHIDEE parameters were adjusted to ITRDB sites while L426 states that also the default parameterization was used.*

This text was revised to avoid confusion. As a summary; five model parameters where manually tuned against 10 ITRDB sites when developing the model functionality to simulate tree ring width (which was well before this study started). These parameters are either global (apply to all forests irrespective of their PFT and location) or PFT-specific (irrespective of their location). Since the start of this study no additional parameters were tuned, all the results are thus based on the default parameter settings.

*Table 2 shows an improvement or deterioration of benchmark metrics if optimization parameters are applied - with mixed results and for few sites -, but from the discussion of these results it remains largely unclear what this really implies for being able to use ITRDB for the proposed purpose [continued]*

The section explaining the verification test was edited for clarity. We refrained from using the term "optimization" because the latest comments made us realize that this caused confusion (L394-399 and L408-425). L384 (L375 in the previous manuscript) was modified for consistency in the model parameter description. We elaborate on the implication around lines L546-550 and L556-567.

*[continued] including information on how representative the experiments in this study are for ITRDB*

Because of the natural variation in forests and site conditions, it is near to impossible to show applicability based on a proof of concept (which is given in this study). Nevertheless, we added Fig. S1 in which we show which domain in climate space was tested in this study. We compare the climate space of the study domain (i.e. the European biomass network) with the climate space of the ITRDB and the climate space of forest biomes. This new figure is referred to in the text around lines 67, 358, and 566.

Section 4.1: The counts and corresponding percentages of improved metrics with optimization appear not to agree with the bold boxes in Table 2. Please check and/or clarify.

Thanks for noticing. This has now been corrected in L433 and 435. Also, bolding in Table 2 was reversed to highlight the cases in which the verification scheme was consistent between the two datasets.

Discussion: Subheadings may aide in better following the presentation.

Done.

Please check captions of supplementary figures, i.e. Fig. S2-S4 show diameter.

We have checked the captions and confirm that they are correct. These graphs show indeed diameter instead of tree-ring width. Young tree and mature tree benchmarks use diameter growth (which equals tree ring width).

Regarding the clarity of the manuscript, i.e. the modelling, data processing and optimizations, it may be helpful to include a diagram of the study design and workflows

We added a figure that depicts the workflow and shows the dependencies between the different parts of the background and the method section (see Fig. 2).

I recommend updating the code repository, especially if this may aide in reproducing your experiments during further review.

The repository was cleaned and updated.

---

## Editor Decision (ED2)

**Topical editor comments on gmd-2020-29**

- L26: ".. have been a corner stone of the Assessment Reports ..."

- L28: please name the four submodels in brackets to make sure non-experts understand

- L129: Please reorder 2,1,3,4 to keep consistency.

- L248: Check reference to Fig. 7

- L424ff: check use of "potentially"

- L503: word missing?

- L515/516: Check sentence structure

- Fig. 1: "Section 3.4" should be "Section 3.3". "Section 4" may be more appropriate than "Section 4.1" specifically.

- Table 1, columns for "Extreme growth": As the metrics are based on RMSE, this should be stated for consistency across the table columns. In addition "scaled" is not a unit and "Extreme growth" in line two should refer to "Timing". Please edit accordingly and add further information in the caption if required.

- Fig. S3: Explain line types in caption or add legend.

- Code and data availability: The specific version of the code and data related to the manuscript should be stored in a frozen repository, which Github does not qualify for. See here for suggestions: https://www.geoscientific-model-development.net/policies/code_and_data_policy.html#item3

- Conflicts of interests: Should include "Philippe Peylin is a member of the editorial board of the journal." See https://www.geoscientific-model-development.net/policies/competing_interests_policy.html

---

## Author Response (AR3)

Dear editor,

We thank you and the referee for the comments to improve the manuscript. As it appeared to us that the main remaining concern for accepting the manuscript is its clarity, we made an effort to refining its structure and explanations (point-by-point response is below) and asked, as suggested by the editor, another colleague, i.e., Valerie Daux, for a pre-review. Although Valerie's work is mostly on the data-side of tree-ring research, she is well aware of the strengths and weaknesses of land surface modelling. We asked Valerie to comment on the manuscript as if she was a referee. Valerie appreciated the nuanced usage of both data and models in the study and considered the proposed benchmarks as possible ways to make some progress in the field. She made several suggestions and recommendations to further improve the flow and clarity of the text.

We feel confident that the flow and clarity of the manuscript has improved and that the manuscript can give new ideas to the land surface modelling community to constrain their models with a long-term benchmark, i.e., tree-ring width.

Kind regards,
Jina Jeong on behalf of the author team

**Referee**
* * *
**Major specific comments:**
(1) Line 206 "virtual forest"
No definitions for the "virtual forest". As authors mentioned in the line 229-230, the proposed method largely relies on the concept of "virtual forest". So, missing its definition is ridiculous.
We added further descriptions about virtual trees in L175-185. Furthermore we realized that the separated numbering for virtual trees and benchmarks which make use of virtual trees was causing a discontinuity in the text flow possibly resulting in confusion. Accordingly, those parts were combined into one section (Section 2.3).

(2) Line 216-220 "Calendar year aligned virtual forest"
I strongly doubted this can be a metric to be compared with outputs of LSMs, because it should have systematic and strong bias as follows.
At the early stage of year, it has low values, because sampled trees (old and big trees) are in its young stages. During the middle period of the data, new trees (at young stage) add to the average calculation, and these trees function as a burden to increase the metric value. During the late period of the data, no tree enters to the average calculation anymore, so the metric value should show intense increasing trend.
The referee is right. Because of this issue during the early stage, the first decades were not used in the analysis which was described in the section for benchmarks (Section 2.3 in the previous revision) but not in the section for virtual trees (Section 2.2 in the previous revision). As mentioned above, we combined descriptions for the virtual trees and benchmark to overcome such discontinuities (Section 2.3).

(3) Lines 141-142
This sentence concludes the paragraph. But, I cannot understand why this conclusion comes out.

We rewrote this sentence for increased readability. (L114-115)

(4) Lines 144-153
Is it possible to rewrite this paragraph so that readers, who are not familiar with dendrochronology and LSMs, can easily understand? Honestly, I still do not understand the point of this paragraph.
More information was added in this part. (L118-125)

(5) Line 241
Here, the "Size related diameter growth" is identical with the "Age-aligned TRWs" in line 212? Inexplicit rephrasing causes confusion, and hence should be avoided.
"Size related diameter growth" signifies that the tree diameter increment tends to decrease as the size of the tree increases, which in turn, forms decreasing tree-ring trends as Fig. 3a and Fig. 4a. "Age-aligned TRWs" describes the alignment of tree-ring widths observation (the difference between Fig. 3 b and c). This inquiry and misunderstanding from the referee may find its origin in the description of virtual trees, therefore, we improved the explanatory part of virtual trees (e.g., L175-185 and L193-196 ).

(6) Lines 253-274
I cannot follow logics here.
We added further descriptions for better understanding (L205 and L211-215).

**Minor specific comments:**
(1) Lines 200-202
Citing figure 4 in these sentences does not make sense.
Figure 3 (figure 4 in the previous revision) shows different virtual trees but also shows how the ITRDB dataset is usually organized. We think this figure can help readers who are not used to tree-ring datasets understand the data. In this revision, as mentioned above, we tried to improve the description of virtual trees (Section 2.3).

(2) Figure 1
I think this figure is needless.
This figure describes the motivation of the study. We agree this is not necessary for the content, so we moved it to the supplementary materials.

(3) Figure 4a
What the vertical axis means? Tree-ring width of the outmost stratum?
As it is now written in the figure, it means tree-ring width for both observation and simulations.

(4) Figure 4c
According to the figure caption, lines on this figure should be identical to those of the figure 4b; Figure 4b and 4c only differ in the alignment on the x-axis. But, shape of the black-dottedlines differ between these figures.
The confusion occurred since black dotted lines in figure 3 (figure 4 in the previous revision) were referred to as a 'virtual tree' without specifying which virtual trees are. We improved the caption of the figure.

(5) Line 676
A typo exists. Location of the period would be immediately after "(c)".

Thanks for noticing. This typo was corrected.

Is this sentence an explanation for the figure 5b? But, figure 5b shows extraction of major
lines on the figure 5a, not the root-mean-square error.
The specifications for corresponding subfigures were added.

**Editor**
* * *
1) One obstacle to the intelligibility may be the enumerations in sections 2.1-2.3 (and beyond), all of
which have 3-5 bullets that partly relate but are not fully consistent, which also raised questions by
reviewer 3. Condensing and/or restructuring section 2 would certainly improve comprehensibility.
We agree with the editor's comment about complexity coming from different numbers of bullets,
therefore, the content for virtual trees and benchmarks were combined (Section 2.3). This reduced the
number of bullet lists. We hope this improved readability.

 The new Fig. 2 connects these elements to some extent but foremost serves as a visual TOC. An
option may be to extend Fig. 3 with the corresponding virtual trees, metrics, processing, etc. if
feasible without overloading.
Fig. 1 (Fig. 2 in the previous revision) was extended to cover virtual trees.

In addition, section 2.3 (iv) contains two metrics, amplitude and timing, which further complicates
connecting the descriptions with display items and results. Please consider these as exemplary notes
and suggestions.
Lines emphasizing the difference of metrics were added in L351 and L472-473.

2) Due to the importance of the virtual tree concept in the study, introducing a separate sub-section
would aid in highlighting this.
We added further descriptions about virtual trees in L175-185. Furthermore we realized that the
separated numbering for virtual trees and benchmarks which make use of virtual trees was causing a
discontinuity in the text flow possibly resulting in confusion. Accordingly, those parts were combined
into one section (Section 2.3).

3) As this study should be considered a proof of concept and presents comprehensive conceptual
elaborations, mentioning this in the abstract would aide in guiding readers' expectations. Also
mentioning the number of sites used for testing would be helpful in this respect.
This context was added to the abstract (L7-8).

4) Legends would improve accessibility of figures. Also check use of "dotted" or "dashed" when
describing line types.
Legends were refined and added to the appropriate figures, i.e., Fig 3,4,5,6 and 7.

Non-public comments to the Author:
I also encourage the authors to have the revised manuscript critically pre-reviewed by colleagues not
involved in the research.

Following the above suggestion, we asked for a pre-review, and the comments from the pre-review were addressed before the submission (see also the cover letter itself).

---

## Editor Decision (ED3)

Editorial comments to authors

1) L38: Reference to section 6 seems to be an artefact. Please replace or remove.

2) L315: Please expand section title, e.g. to "Verification of benchmarks based on big-tree data" to avoid confusion with verification of model simulations that are described in the preceding sections.

---

## Author Response (AR4)

Dear Chris Folberth,

Thank you for your careful reading and comments on our manuscript. Following your remaining concerns, we substantially and carefully revised the logic, structure, and terminologies used in the manuscript. Virtual trees, the construction of observational benchmarks from these, and the comparison with land surface simulations via the two metrics for each benchmark are, we hope, now much clearer to the reader. This resulted in substantial changes throughout the entire manuscript and its supplementary material. All sections were restructured and revised for logic, clarity. References in text to specific results in the tables, figures, and supplement in support of the discussion have been added. Figures 4-8, clarifying the construction of virtual tree benchmarks and the estimation of skill metrics, were revised; all figure captions were reviewed for clarity and accuracy. The changes in the results table you spotted occurred when we transferred the document from Word to LaTex; we regret this error, and are thankful you noticed the mistake for us to correct in revision. All reported results have been rechecked against the raw data to ensure that the transfer between text editors did not result in other errors.

We hope this revision addresses your remaining concerns with the manuscript. Thank you for your consideration.

Kind regards,
Jina Jeong on behalf of the author team
* * *
1) L80: Reference to Figure 1: In Fig. 1 you first mention the virtual trees, which have so far not been defined. The caption needs to ensure that the figure can be understood. Add a brief statement on the virtual tree concept and where more information can be found in the manuscript.
We agree. The information was added in short to the caption of Fig. 1.

In addition, the term "maximum virtual tree" is not explained anywhere and only used in Fig. 3 besides here.
We agree. All 'maximum virtual tree' in the text have changed to 'largest virtual tree'.

2) L145-L170: Same issue as above: Fig. 3 is referenced, which requires understanding the concept of virtual trees but virtual trees have not been introduced to the reader and there is no information where their definition can be obtained.

Following your comment we realized that Fig. 3 served too many roles at the same time. This issue was solved by drawing a new figure (Fig. 3) to give an example of ITRDB datasets and referring to Fig. 4 only to explain the challenges and solutions.

3) Caption of Fig. 3: "maximum virtual tree" not clear.

As mentioned earlier, all changed to 'largest virtual tree'.

4) Section 2.2: I recommend to insert a separate (sub-)section on the virtual trees or at least mention the term in the section title, which would make it easier to point readers to this. I.e., rename section 2.2 to e.g. "...and addressing them using a concept of virtual trees" and/or split this section into (a) challenges and (b) the solutions proposed in this manuscript.

We followed this recommendation. The section for "Challenges of using ITRDB data as a long-term benchmark" (section 2.2) was separated from the section "Solutions for the challenges and virtual trees". Furthermore a bullet list was used to enhance the presentation of the virtual trees.

5) L201/L216 etc.: remove "across many sites". An ideal model would have this behavior across all sites.

All occurrences were removed (L207, L222, L234, and L251).

6) Section 2.3 (ii) and Fig. 5: How is the cut-off time period (last ten decades) determined? Section 2.3 (iii) has a thirty year time period and it is mentioned that this may be arbitrary.

It is arbitrary as described in section 2.4 (iii). Text was added to explain that this threshold may vary for different regions in the world (L218-221).

7) L373: I don't understand the reference to Fig. 9 here. Wouldn't Tab. 1 be more appropriate to show the 88 test cases?

The figure was suggested by the referee during the second revision to show the benchmark examples with European biomass network datasets. The figure has now been moved to Supplementary.

8) L380: 72% is not correct anymore as apparently results for site SOB have changed from 0.001 to -0.001 since the last version of the manuscript, which would increase the rate to 77%.

Thank you for noticing this. It was a mistake that happened during changing platforms. The sign was corrected. And after further checks against the raw data, some additional small inconsistencies in the table were found. All were corrected during this revision.

9) L386: Aren't there additional sites that have a positive change?

You are right. Those additional sites (HD2, SOB, and SOR) showed only marginal positive differences therefore there was no further explanation. A sentence was added (L385-386) to clarify the issue.

10) Figure 7 legend: "Simulated biggest output" not clear

We could not find the corresponding phrase in the caption, but 'biggest class' in Fig 5 to 7 was changed to 'largest diameter class' for consistency.

11) Supplement L348: Table S3 does not provide information on "issues of the ITRDB"

You are right. That reference was removed.

12) Supplement L384: "longer" -> "older"?

applied.

13) Supplement L388: "XX mm"?

It was fixed to 1 mm.

14) Supplement L412: "biggest simulation" not clear -> "largest simulated diameter class"?

It was changed following your suggestion.

---

## Author Response (AR5)

Dear Chris Folberth,

We are grateful for the time you have taken to review the subsequent version of our manuscript. We have addressed all the technical corrections in the main text and supplementary materials following your comments. The code repository was archived on zenodo with an assigned doi. The conflicts of interests section was modified following the journal's rule.

Thank you for your help to improve the manuscript.

Kind regards,
Jina Jeong on behalf of the author team